# Ambiguous Strategic Classification

**Ivri Hikri** [1]   **Nir Rosenfeld** [1]

## Abstract

A common assumption in strategic classification is that the classifier is public knowledge. However, it remains unclear whether, and why, a system would choose to commit to full disclosure. We study a setting in which regulation requires the system to disclose some, but not all, of the information. This induces a learning task in which the learner must jointly optimize the classifier and the uncertainty surrounding it. To this end, we adopt from robust mechanism design the notion of *ambiguity*, which in our setting allows the learner to reveal a set or range of possible classifiers, while privately choosing which of them to ultimately realize. We investigate how ambiguity affects the learning task, develop efficient algorithms for computing best-responses and training, and empirically explore strategic learning and its outcomes in this novel setting and using our approach.

## 1. Introduction

Strategic classification studies learning in a setting where users can modify their features to obtain favorable predictive outcomes (Brückner et al., 2012; Hardt et al., 2016). This applies to domains such as loan approval, university admissions, job hiring, social welfare benefits, medical program eligibility, and insurance claims. The original problem formulation cleanly captures a form of tension that can arise between a learning system and its users, but relies on several key assumptions, one of which is that users know the classifier exactly. This assumption enables the system to anticipate how users will respond to any choice of classifier, and as a result, preemptively train a classifier to be strategically robust. Thus, and perhaps surprisingly, granting users full access to information regarding the classifier can be beneficial to the system (Ghalme et al., 2021; Bechavod

[1]Faculty of Computer Science, Technion – Israel Institute of Technology, Haifa, Israel. Correspondence to: Nir Rosenfeld <nirr@cs.technion.ac.il>.

*Proceedings of the $43^{rd}$ International Conference on Machine Learning*, Seoul, South Korea. PMLR 306, 2026. Copyright 2026 by the author(s).

et al., 2022), this providing a utilitarian justification and motivation for transparency.

Nonetheless, and even though transparency can be helpful, this does not imply that full disclosure is always the optimal strategy. Given that classifier information begins as private to the system, it is generally within the system's power to decide what information to reveal—and what not to. As such, our work aims to understand what happens when a strategic system controls not only the choice of classifier, but also the information regarding the classifier that is made public. By interpolating between the settings of no information and full information, our framework allows to ask when, if, and how learning can exploit user uncertainty to promote its own goals; what implications this may carry for users; and which forms of regulation are required or can be useful.

To model user uncertainty regarding the classifier, we borrow from the field of robust mechanism design, and assume users make decisions under *ambiguity*. Ambiguity captures settings where the realized outcome is one of several predetermined possibilities, but where possible outcomes are not attached to any probabilities. This contrasts the common notion of statistical uncertainty (which has been considered in strategic classification, e.g., Jagadeesan et al. (2021); Cohen et al. (2024b)), and is appropriate for settings where a strategic entity, rather than randomness, determines both the set of possible outcomes and the eventual realization.

Consider for example a firm whose hiring process includes a coding interview. The firm announces that interviews will focus on either problem-solving skills or algorithmic fundamentals, but will include only one of them. Applicants who wish to succeed must therefore prepare for, and possibly improve in, *both* areas. This creates an advantage for more qualified applicants who are already both skilled and knowledgeable—not because only they can pass the interview, but because candidates proficient only in one topic cannot know if they will pass. If improving across both topics proves too costly for such candidates, the hiring process becomes effective due to its inherent ambiguity.

Following the above example, our work operationalizes the idea of ambiguity within the strategic classification framework by assuming that users are given an *ambiguous set* of possible classifiers, e.g., with parameters in some bounded range. Of these, users know that one will realize—but do

not know which one. Consequently, we model users as risk averse, i.e., acting to maximize utility under the worst-case possible outcome (Gilboa & Schmeidler, 1989). The goal of learning in this setting, which we refer to as *ambiguous strategic classification*, is then to maximize accuracy by jointly learning the true classifier and a corresponding ambiguous set, which shapes user responses. To restrict the power of learning to exploit user uncertainty, we introduce a regulator entity that controls the type and degree of ambiguity permitted to the learner. This allows to balance between the learning objective (namely accuracy) and societal goals (e.g., social welfare or burden).

Since ambiguity protects system information by restricting user access, it is tempting to assume that higher ambiguity is useful to the system and harmful to its users. However, one of our conclusions is that outcomes are best under the "right" amount of ambiguity. Hence, the main conceptual challenge of learning is to correctly balance between what information is revealed, and what is kept private, in terms of how this affects learning outcomes through downstream user behavior.

The technical challenges of learning in this setting are twofold. First, ambiguity gives rise to user behavior in the form of max-min best responses. Even for the common choice of linear classifiers and 2-norm costs, these can be challenging to solve. Nonetheless, and for three distinct and increasingly complex ambiguity types, we show how best responses can be computed efficiently by exploiting the structure of the learning task. These are enabled by reduction to appropriate and tractable projection problems. Second, under ambiguity, standard losses (e.g., log loss) are deemed inefficient, and appropriate strategic losses (e.g., the strategic hinge) become intractable. For this, we devise a relaxation of the generalized strategic hinge that, coupled with our efficient projection algorithms, provides a streamlined differentiable loss suitable for ambiguous learning tasks.

We conclude with a series of experiments that explore our setting of ambiguous strategic learning. Using both synthetic and real data coupled with simulated user behavior, we demonstrate how our approach can find effective ambiguous classifiers, and examine the effects of regulation on learning and its outcomes for both system and the user population.

### 1.1. Related work

**Strategic classification.** The field of strategic classification has gained substantial traction since its introduction (Brückner et al., 2012; Hardt et al., 2016). Many works have since aimed to extend the original framework, in particular by supporting different notions of uncertainty. One line of research considers learning under fixed but unknown costs, including in the online (Dong et al., 2018; Ahmadi et al., 2021), multi-round batch (Lechner et al., 2023), and one-shot (Rosenfeld & Rosenfeld, 2024) settings; under personalized costs

(Lechner et al., 2023; Shao et al., 2024); and for general manipulation graphs (Ahmadi et al., 2023; Cohen et al., 2024a). Another thread focuses on uncertainty regarding the environment, such as through causality (Miller et al., 2020; Harris et al., 2022; Horowitz & Rosenfeld, 2023), feedback (Harris et al., 2023), or market forces (Sommer et al., 2025).

Closer to ours are works that model uncertainty regarding the classifier, either as missing information (Ghalme et al., 2021; Bechavod et al., 2022; Barsotti et al., 2022) or due to noise (Jagadeesan et al., 2021; Geary & Gouk, 2025). In particular, Ghalme et al. (2021) make the point that user-side uncertainty can entail system-side uncertainty regarding best responses. This raises questions of how to contend with private user information (Levanon & Rosenfeld, 2022), whether and when systems should release their classifier (Shao et al., 2025), and how the system can benefit by controlling user uncertainty (Cohen et al., 2024b). The latter's setting is perhaps closest to ours, but is distinct in that: (i) the system adopts a Bayesian persuasion perspective, (ii) both users and the system rely on a joint prior, (iii) users maximize expected utility w.r.t. this prior; and (iv) uncertainty sets are constructed over a given, fixed classifier (i.e., there is no learning). In contrast, we consider worst-case ambiguity rather than statistical uncertainty, do not require priors or shared information, and focus predominantly on learning.

**Ambiguity and robustness.** Attaining robustness through worst-case modeling is central to both machine learning and economics. In learning, minimax objectives are common for attaining e.g. adversarial robustness (Goodfellow et al., 2015) and distributional robustness (Sinha et al., 2018). Within strategic learning, this perspective has been adopted to contend with strategic feature selection (Nair et al., 2022) or unknown costs (Levanon & Rosenfeld, 2021), and to bridge strategic and adversarial training (Ehrenberg et al., 2025). Economics makes a distinction between uncertainty with attached probabilities, namely risk, and without them, known as Knightian uncertainty (Knight, 1921). Within mechanism design, ambiguity captures settings where such uncertainty, as perceived by one agent, can be strategically shaped by another. The goal is then to design mechanisms that are robust to or that can leverage such uncertainty (Bergemann & Morris, 2005). This general idea has been used for designing robust auctions (Bergemann et al., 2016), contracts (Carroll, 2015), persuasion schemes (Dworczak & Pavan, 2022), and markets (Rigotti et al., 2008), as some examples. Our paper introduces the notion of ambiguity robustness into strategic classification.

## 2. Setting

Our setup builds on conventional strategic classification in the batch setting (Hardt et al., 2016; Levanon & Rosenfeld, 2021) and generalizes it to support ambiguity. Users

are described by features $x \in \mathcal{X} = \mathbb{R}^d$ and binary labels $y \in \mathcal{Y} = \{\pm 1\}$, with pairs $(x, y)$ sampled iid from some unknown joint distribution $D$. The primary goal of learning is to find a classifier $h$ from a chosen class $H$ that makes accurate predictions, $h(x) = \hat{y} \approx y$. The challenge is that once $h$ is deployed, users respond by strategically modifying their features $x \mapsto x^h$ if this secures them a positive prediction, and is cost effective. Given a cost function $c(x, x')$ that governs modification costs, users *best-respond* as:

$$x^h = \Delta_h(x) \triangleq \operatorname*{argmax}_{x'} h(x') - \alpha c(x, x') \qquad (1)$$

where $\alpha > 0$ determines the tradeoff between the utility users gain from a positive prediction, namely $h(x')$, and the incurred cost $c(x, x')$. Our best response algorithms in Sec. 4 apply to any convex cost. Our learning algorithm in Sec. 5 applies to a broad family of costs of the form $\phi(\|A^{1/2}(x - x')\|_p)$ where $\| \cdot \|_p$ is any $p$-norm ($p \geq 1$), $\phi$ is any non decreasing function, and $A$ is any PD matrix (Rosenfeld & Rosenfeld, 2024). For simplicity we focus throughout mostly on the common special case of 2-norm costs, $c(x, x') = \|x - x'\|_2$, in which points can move to a distance of at most $2/\alpha$ in all directions. We comment on extending to the generalized case where useful, and defer its full treatment to the Appendix.

The goal of learning is to find a strategically robust classifier, this by aiming to maximize *expected strategic accuracy*:

$$\operatorname*{argmax}_{h \in H} \mathbb{E}_D[\mathbb{1}\{h(\Delta_h(x)) = y\}] \qquad (2)$$

In practice, this is achieved by optimizing an empirical surrogate objective over a training set $S = \{(x_i, y_i)\}_{i=1}^m$ sampled iid from $D$ and using a proxy loss (e.g., hinge).

**Ambiguity.** Standard strategic classification assumes that users know $h$ exactly-as implied in Eq. (1). We generalize this to support users acting under ambiguity regarding which classifier will be used. Formally, let $\Gamma \subseteq H$ be a set of possible classifiers, which we refer to as the *ambiguity set*. We will assume that ambiguous sets are always truthful in that they must include the true $h$. Our main modeling assumption is that users are risk averse: when facing ambiguity, they will act to maximize utility under the worst-case outcome. This gives rise to the *max-min best-response* mapping:[1]

$$\Delta^\Gamma(x) \triangleq \operatorname*{argmax}_{x'} \min_{h \in \Gamma} h(x') - \alpha c(x, x') \qquad (3)$$

---

[1]A plausible alternative is to model users as Bayesian agents who maximize expected utility under probabilistic beliefs. Such beliefs constitute private information that the learner cannot observe, and therefore cannot anticipate without additional assumptions or mechanisms (e.g., elicitation). In contrast, we consider settings in which the learner controls the uncertainty itself. Our approach is appropriate when users are willing to invest effort to guarantee a positive prediction—rather than put faith in favorable odds.

Note that setting $\Gamma = \{h\}$ recovers the standard strategic classification setting (Eq. (1)), whereas $\Gamma = \mathcal{Y}^\mathcal{X}$ reverts to the non-strategic setting since all responses are suppressed. We refer to all $h' \in H$ as *possible* classifiers, and distinguish between *realized* (or *true*) $h$ that will be used in practice and which determines effective user utility, and all other *non-realized* $h' \neq h$ that only affect user behavior via Eq. (3).

**Ambiguity sets.** We consider several types of ambiguity sets of increasing structural complexity. For concreteness, and as is common in the strategic learning literature, we focus on linear classifiers as a base class, $H_{\mathrm{lin}} = \{h_{w,b}(x) = \mathrm{sign}(w^\top x + b) : w \in \mathbb{R}^d, b \in \mathbb{R}\}$. Nonetheless, and as we will see, ambiguity introduces nonlinearities that act to increase the effective model class capacity.

As a motivating example, consider the task of designing a qualifying exam, e.g., academic or for job hiring. Features represent answer correctness on questions of different topics or subjects, and weights represent the relative importance of each topic to the total score. The types are:

- **Offset**: Fixed known weights $w \in \mathbb{R}^d$, but an ambiguous offset $b$ in a given range: $\Gamma = \{h_{w,b} : b \in [\underline{b}, \bar{b}]\}$. This corresponds to determining in advance the importance of each topic via a fixed score function, but keeping the acceptance threshold ambiguous.

- **Discrete**: A set of $k$ distinct classifiers, $\Gamma = \{h_1, \ldots, h_k\}$, where $h_j = h_{w_{(j)}, b_{(j)}}$ for each $j \in [k]$. This corresponds to releasing a set of possible exams, while committing to eventually using one of these.

- **Continuous**: Ambiguous weights $w$ with bounded entries $w_i \in [\underline{w}_i, \bar{w}_i]$, and shared $b$: $\Gamma = \{h_{w,b} : w \in [\underline{w}, \bar{w}]\}$.[2] This corresponds to stating for each topic a range for its possible importance in the final weighted sum.

We will be interested in understanding how the type and degree of ambiguity affects learning and its outcomes, as determined by the type and size of $\Gamma$.

**Controlling ambiguity.** A central question is who determines the ambiguity users face, and how. If $\Gamma$ is fixed and predetermined, e.g. by a regulator, then the problem reduces to learning under $\Gamma = H$, which means that users act 'in the dark' (akin to Ghalme et al. (2021)). At the other extreme, if the learner has complete control over $\Gamma$, then it can entirely suppress strategic behavior—for example, by setting $\Gamma = \{h, 1 - h\}$ (though as we argue, this may not be optimal). The more interesting settings, and our focus, lie between these extremes: the regulator imposes *restrictions* on $\Gamma$, while the learner remains free to choose $\Gamma$ from within the feasible set. For example, the regulator may con-

---

[2]Our results also support continuous ambiguity in both $w$ and $b$, enabled by replacing $b$ with $\underline{b}$ throughout. For clarity, and since it provides only mild further expressive power, we focus on continuous ambiguity in $w$, and treat offset ambiguity independently.

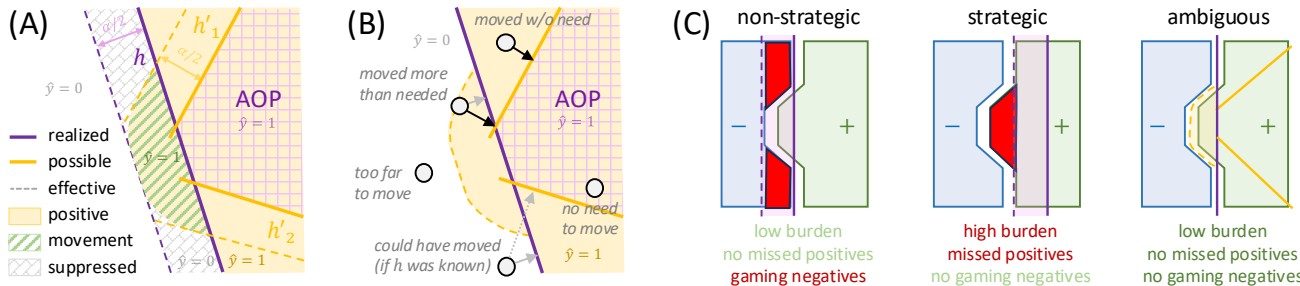

*Figure 1.* **(A)** The ambiguous strategic setup, demonstrated for discrete ambiguity with $\Gamma = \{h, h'_1, h'_2\}$. Points move only if they can reach the AoP (purple). Adding possible classifiers $h'$ suppresses movement (grey) by shrinking the AoP and consequently the movement region (green). Points are $x$ classified as positive (yellow) if either $h(x) = 1$ or they can move. **(B)** Ambiguity gives rise to more nuanced strategic behavior and outcomes. **(C)** Simple example comparing optimal non-strategic, strategic, and ambiguous strategic classifiers. For the non-strategic classifier, movement (purple region) enables gaming behavior. For the strategic classifier, movement (purple region) induces burden, and linearity entails errors. The ambiguous strategic classifier (purple line) leverages ambiguity (yellow lines) to express a non-linear effective boundary that separates that data and entails low social burden for points that move (yellow region).

straint the size of $\Gamma$ (e.g., fix $k$ for discrete, or upper-bound $\|\bar{w} - \underline{w}\|_\infty$ for continuous), or require all possible $h' \in \Gamma$ to be similar to the true $h$ (e.g., in parameters or predictions).

**Social burden.** Following Milli et al. (2019), we use *social burden* to measure the negative impact of strategic learning on the user population. This is defined as the average cost needed for positive users to gain positive predictions. Note that these are theoretical costs which can exceed $2/\alpha$. The natural generalization to our setting is:

$$\text{burden}(\Gamma) = \frac{1}{|\{i : y_i = 1\}|} \sum_{i:y_i=1} \min_{x' : \forall h \in \Gamma, h(x')=1} c(x_i, x')$$

which captures how much positive users will pay as risk averse to guarantee a positive prediction given $\Gamma$.

## 3. Preliminaries

We begin with several basic properties that will help gain intuition and be useful downstream. Proofs in Appendix A.

### 3.1. How points move

The key distinction of our setting is that users move only if this guarantees them a positive prediction by *all* possible classifiers. This idea is captured by the following definition.

**Definition 1** (AoP)**.** For a given $\Gamma$, its *area of positivity* is:

$$\mathbb{P}(\Gamma) = \{x \in \mathcal{X} : h'(x) = 1 \ \forall h' \in \Gamma\} \qquad (4)$$

i.e., the set of points classified as positive by all possible $h'$.

Since users maximize for worst-case outcomes, it will be worthwhile for a point to move only if the AoP is within its reach. Thus, the *induced movement region* is defined as the set of all points $x$ that are outside the AoP but at distance (i.e., cost) at most $2/\alpha$ from it. Further, since users

minimize costs, movements are always onto the boundary of the AoP. These are illustrated in Fig. 1 (A).

**Non-linearity via (linear) ambiguity.** When $\Gamma$ comprises linear classifiers, $\mathbb{P}(\Gamma)$ is by definition a (convex) intersection of halfspaces. Thus, while the classifier itself remains linear, ambiguity introduces non-linearities into how users respond. This in turn gives learning the capacity to express effective non-linear decision boundaries indirectly by influencing user behavior.

Whereas in standard strategic classification (i.e., $\Gamma = \{h\}$) movement and classification outcomes are coupled, under ambiguity, this connection breaks. In particular, ambiguity works to suppress movement: some points $x$ that could have attained a positive prediction had they known the true $h$ may not move under ambiguity when it is strictly more demanding to satisfy all $h' \in \Gamma$. Meanwhile, other points may move—and pay the cost—unnecessarily. We refer to the difference between $\mathbb{P}(\{h\})$ and $\mathbb{P}(\Gamma)$ as the *suppressed region*, and it is the non-linearity of this region that shapes outcomes. These ideas are illustrated in Fig. 1 (B).

### 3.2. What ambiguity does

The main mechanism through which ambiguity affects outcomes is by controlling the size and shape of the AoP. This effect is monotone in the following sense.

**Observation 1.** *Let $\Gamma$ be an ambiguity set, then for any larger $\Gamma' \supseteq \Gamma$, its corresponding AoP can only be smaller, i.e., $\mathbb{P}(\Gamma') \leq \mathbb{P}(\Gamma)$.*

This follows directly from the monotonicity of intersection. Note that in terms of accuracy, a larger AoP is neither 'good' nor 'bad'; rather, what matters is the class ratio of points for which movement is suppressed. This is since accuracy improves when points with $y = 0$ are driven out of the induced region (since $\hat{y} = 0$), but degrades when such points

have $y = 1$ (since no longer $\hat{y} = 1$). Learning can therefore contend with (or exploit) ambiguity by finding (linear) $h$ for which the (non-linear) suppressed region due to $\mathbb{P}(\Gamma)$ includes more negatives than positives. Note that in general, the optimal $\Gamma$ will neither be $\{h\}$ (full information) nor $\mathcal{Y}^{\mathcal{X}}$ (full ambiguity), but rather, should encode the 'right' type and amount of ambiguity. This is demonstrated in Fig. 1 (B).

**Accuracy.** For a given $D$ and $h$, denote by $\mathrm{acc}(h)$ the standard (non-strategic) expected accuracy, and by $\mathrm{acc}(h; \Delta)$ the expected accuracy when users best respond via $\Delta$. A known result is that in standard strategic classification user behavior can only degrade performance: for any $D$ and any $h$, $\mathrm{acc}(h; \Delta_h) \leq \mathrm{acc}(h)$. Furthermore, both attain the same theoretical optimum, namely $\max_h \mathrm{acc}(h; \Delta_h) = \max_h \mathrm{acc}(h)$ (Rosenfeld & Rosenfeld, 2024). Interestingly, ambiguity can help improve accuracy *beyond* what is possible absent strategic movement. That is, and as we will see, in certain settings $\max_h \mathrm{acc}(h; \Delta_h^{\Gamma}) > \max_h \mathrm{acc}(h)$. In fact, ambiguity can make non-separable data become separable, this by using $\Gamma$ to "linearize" the data in the shifted induced distribution. Fig. 1 (C) demonstrates this idea on a simple example with discrete ambiguity using $k = 3$.

**Social burden.** Since increased ambiguity shrinks the AoP, the cost of reaching it can only rise for users. This has social implications; a direct result of Lemma 1 is:

**Corollary 1.** *If $\Gamma' \supseteq \Gamma$ then $\mathrm{burden}(\Gamma') \geq \mathrm{burden}(\Gamma)$.*

Importantly, this should not be taken to imply that *learning* with looser ambiguity constraints necessarily increases social burden; as we will see, the opposite is quite plausible.

### 3.3. Case study: Offset ambiguity

Consider first an offset ambiguity task where and the goal is to learn $w$ and $\underline{b}, \bar{b} \in \mathbb{R}$, possibly under some constraints, and where users respond to $\Gamma = \{h_{w,b} : b \in [\underline{b}, \bar{b}]\}$. How does ambiguity affect learning outcomes in this case? Our next result shows offset ambiguity has only limited effect.

**Lemma 1.** *Strategic classification with offset ambiguity is equivalent to standard strategic classification with a modified cost scale $\alpha \leq \alpha_\Gamma \leq \alpha(1 - \frac{\bar{b}-\underline{b}}{2\|w\|_2}\alpha)^{-1}$.*

That is, Eq. (3) reduces to Eq. (1) but with $\alpha_\Gamma$ instead of $\alpha$. This is since the AoP remains linear, and the movement region becomes a band of width $2/\alpha$ on the negative side of $h_{w,\underline{b}}$, rather than of the true $h_{w,b}$. The effective value of $\alpha_\Gamma$ is determined by the choice of the true $b$ in the range $[\underline{b}, \bar{b}]$. As a result, offset ambiguity cannot help improve the maximal attainable accuracy (beyond the non-strategic case), but can suppress movement further, and therefore can potentially improve the accuracy achieved in practice.

## 4. Computing best responses

The first step to strategic learning is computing best responses. For standard, fully informed strategic classification, standard best responses can be easily computed (see below). However, under ambiguity, max-min best-responses are no longer straightforward to solve. We next show how such best responses can nonetheless be computed efficiently for the types of ambiguity we consider. These results will be key to our learning approach in Sec. 5. We defer all proofs to Appendix A.

**Best responses projections.** In the full information setting, computing $\Delta_h(x)$ from Eq. (1) can be done in three steps: (1) check if $h(x) = 0$; if so, (2) compute $x^*$ as the projection of $x$ onto $h$, and (3) modify $x \mapsto x^*$ only if $c(x, x^*) \leq 2/\alpha$; otherwise, keep $x$ unchanged. When $h = h_{w,b}$ is linear and $c$ is 2-norm (or quadratic), the projection admits a simple closed form solution: $x^* = x - \frac{w^\top x + b}{\|w\|}w$. Our main observation here is that this general procedure also applies to max-min responses $\Delta_h^\Gamma$ from Eq. (3), but with the adjustment that conditional projections are made onto the AoP:

$$x^* = \mathrm{argmin}_{x'} \|x - x'\|_2 \quad \text{s.t.} \ x' \in \mathbb{P}(\Gamma) \quad (5)$$

which generalizes the standard case of $\Gamma = \{h\}$. We next discuss how to compute Eq. (5) for each ambiguity type.

### 4.1. Offset and discrete ambiguity

**The offset case.** Let $\Gamma = \{h_{w,b} : b \in [\underline{b}, \bar{b}]\}$ for some $w$ and $\underline{b}, \bar{b} \in \mathbb{R}$ with $\underline{b} \leq \bar{b}$. Due to Lemma 1, the AoP remains a hyperplane, namely that defined by $w$ and $\underline{b}$. Under 2-norm costs, projections are also linear, and hence similar to the standard case, but using $\underline{b}$ instead of $b$. The effect of $\bar{b}$ on outcomes comes through the need to modify the cost threshold for modification to be $2/\alpha_\Gamma$, where $\alpha_\Gamma = \alpha(1 - \frac{\bar{b}-\underline{b}}{2\|w\|_2}\alpha)^{-1}$.

**The discrete case.** For a discrete $\Gamma = \{h_1, \ldots, h_k\}$, and for convex costs $c(x, x')$, a projection $x^*$ can be formulated as the solution to the convex program with linear constraints:

$$\mathrm{argmin}_{x'} c(x, x') \quad \text{s.t.} \ w_{(j)}^\top x' + b_{(j)} \geq 0 \ \ \forall j \in [k] \quad (6)$$

While there is no closed form solution for this projection, it can be solved efficiently (for a reasonable choice of $k$) using any convex program solver. Note Eq. (6) can be extended to support any class of convex classifiers expressable as tractable constraints.

### 4.2. Continuous ambiguity

Consider now the more challenging continuous case, where ambiguity sets are of the form $\Gamma = \{h_{w,b} : w \in [\underline{w}, \bar{w}]\}$ for some $b \in \mathbb{R}$ and $\underline{w}, \bar{w} \in \mathbb{R}^d$. Though the general template of Eq. (6) still applies, rather than a discrete set, the program

now has a *continuum* of constraints:

$$\text{argmin}_{x'}\ c(x, x')\ \ \text{s.t.}\ \ w^\top x' + b \geq 0\ \ \forall w \in [\underline{w}, \bar{w}] \quad (7)$$

for which there is no generic solution or solver.

Our first step is to construct an equivalent program with a discrete (yet large) constraint set that admits the same solution as Eq. (7). Define the following set of weights:

$$\Omega = \{w \in \mathbb{R}^d : w_i \in \{\underline{w}_i, \bar{w}_i\}\ \forall i \in [d]\} \quad (8)$$

That is, $\Omega$ comprises all "extreme" weight vectors $w$, with each coordinate $w_i$ taken to be either $\underline{w}_i$ or $\bar{w}_i$. Note the size of $\Omega$ is $2^d$. Next, define the program:

$$\text{argmin}_{x'}\ c(x, x')\ \ \text{s.t.}\ \ w^\top x' + b \geq 0\ \ \forall w \in \Omega \quad (9)$$

**Lemma 2.** *The solution to Eq. (9) is optimal for Eq. (7).*

The proof relies on showing that the feasible sets of Eq. (7) and Eq. (9) are equivalent; see Appendix A.1. While Eq. (9) is now admissible for convex solvers, the exponential size of its constraint sets makes its computation (and representation) prohibitive. We next show that it is nonetheless possible to solve it efficiently by providing an appropriate separation oracle, which enables polynomial runtime in theory and linear (and often less) in practice.

**Projection by separation.** A *separation oracle* is a subroutine within a convex solver that for a given non-feasible point $x$ finds a hyperplane that separates it from the feasible set. A known result is that the ellipsoid method coupled with an efficient separation oracle is guaranteed to find the optimal solution in polytime even if the number of constraints is exponential (Grötschel et al., 2012). Our next result identifies such an oracle for Eq. (9).

**Lemma 3.** *Let $\Omega$ as in Eq. (8), and consider some non-feasible $x$. Define $w^*$ with $w_i^* = \bar{w}_i$ if $x_i < 0$, and $\underline{w}_i$ otherwise. Then $w^* \in \Omega$ and separates $x$ from the feasible set defined by $\Omega$ and $b$.*

The resulting separating hyperplane is tight, i.e., corresponds to one of the (exponentially many) constraints. It can be determined efficiently via closed form solution, this by exploiting the fact that the worst $w \in \Omega$ for $x$ is determined by the orthant of $x$. Proof in Appendix A.2.

**Cutting plane algorithm.** The ellipsoid method is theoretically efficient, but in practice can be cumbersome and slow. The common solution is to instead use a cutting plane approach which sequentially adds constraints to an 'active' constraint set, using the separation oracle, until an optimal solution is found. Pseudocode is given in Algo. 1.

**Theorem 1.** *Algorithm 1 solves Eq. (9).*

Proof in Appendix A.3. The algorithm relies on the fact that the oracle finds the 'worst' $w$ for $x$, i.e., having the lowest

---

**Algorithm 1** ProjectContinuousAoP

1: **Input:** example $x$, parameters $\underline{w}, \bar{w}, b$
2: **initialize** constraint set $\mathcal{C} \leftarrow \varnothing$, projection $x^* \leftarrow x$
3: **repeat**
4:     define $w$ with entries $w_i \leftarrow \underline{w}_i$ if $x_i^* \geq 0$ else $\bar{w}_i\ \forall i$
5:     **if** $w^\top x^* + b \geq 0$ **then** break      ▷ *correct $x^*$ found*
6:     $\mathcal{C} \leftarrow \mathcal{C} \cup \{w\}$
7:     $x^* \leftarrow \text{argmin}_{x'}\ c(x, x')$ s.t. $v^\top x' + b \geq 0\ \forall v \in \mathcal{C}$
8: **until** $\mathcal{C} = \Omega$      ▷ *worst case; typically $\sim d$ steps*
9: **return** $x^*$ as projection of $x$ onto $\mathbb{P}(\Gamma_{\underline{w}, \bar{w}, b})$

---

score of all $v \in \Omega$. Thus, as the constraint set $\mathcal{C}$ grows linearly, the number of constraints effectively accounted for can potentially grow much faster. The terminating condition ensures that once $\mathcal{C}$ accounts for all $v \in \Omega$, the correct projection is found. In the worst case, runtime can be exponential; but in practice the number of iterations is linear in $d$, and typically even smaller. This was true both in our experiments in Sec. 6 and in a designated analysis (Appendix D).

Algo. 1 relies on the assumption that the AoP is not empty. The following provides a necessary and sufficient condition:

**Lemma 4.** $\mathbb{P} \neq \varnothing$ *iff $b \geq 0$ or $\underline{w}_i \cdot \bar{w}_i > 0$ for some $i \in [d]$.*

We use this lemma to form constraints for our method in Sec. 5. If the AoP is empty, then $\Delta_x^\varnothing(x) = x$, i.e., points do not move.

## 5. Learning approach

We now turn to present our algorithm for strategic learning under ambiguity. Our general approach will be to devise an appropriate differentiable proxy loss that accounts for strategic user behavior, and optimize a corresponding empirical surrogate objective. Formally, given a model class $H$, regulatory constraints on ambiguity $\mathcal{G}$, and a sample set $S = \{(x_i, y_i)\}_{i=1}^m$, our objective will be of the form:

$$\underset{\Gamma \in \mathcal{G}, h \in \Gamma}{\text{argmin}}\ \frac{1}{m} \sum_{i=1}^m \ell(x, y; h, \Gamma) + \lambda R(w) \quad (10)$$

where $\ell$ is the loss function, $R$ is a regularization term, and $\lambda$ is its coefficient. We will make the assumption that $\mathcal{G}$ requires all possible classifiers to be 'reasonable' alternatives to the true one by constraining weak agreement in the form of $w^\top v > 0$ for all $v \in \Gamma$. This prevents adding to $\Gamma$ classifiers that contradict the true $h$, ensuring that moving in response to any other $h'$ also increases the score w.r.t. $h$, which we view as normatively desirable. Weak agreement will also be technically useful for our approach, which leverages it for correctness. In practice we enforce agreement as a soft constraint via an additional penalty term.

While the main challenge in solving Eq. (10) lies in optimiz-

ing $h$ and $\Gamma \ni h$ jointly under constraints $\mathcal{G}$, an additional modeling challenge is that strategic learning often requires a specialized loss function suitably tailored to the task at hand. To see why, consider the naïve approach of applying a standard loss function $\ell$ (e.g., hinge loss or log-loss) to strategic inputs as $\ell(\Delta_h(x), y; h)$. Because points respond to the classifier by moving onto its decision boundary, any point that moves will obtain a score of zero, and a loss of one. This will remain to hold even if $w, b$ are perturbed, since points will just re-adapt to how the classifier has changed. As a result, gradients will not carry any information regarding these points, and training will be ineffective. Moreover, this approach requires differentiating through the argmax best response mapping $\Delta$, which can be challenging.

**The strategic hinge.** A common choice of loss function for standard strategic classification is the *generalized strategic hinge* (Levanon & Rosenfeld, 2022), which gives a generic formula for implementing strategic losses in diverse settings. For the special case of linear classifiers and 2-norm costs, the strategic hinge loss takes the following simple form:

$$\ell_{\text{strat}}(x, y; h) = \max\{0, 1 - y(w^\top x + b + \frac{2}{\alpha}\|w\|_2)\} \quad (11)$$

which is differentiable (though not necessarily convex), and does not include $\Delta$ explicitly. Note that this uses $y \in \{\pm 1\}$ rather than $\{0, 1\}$. Unfortunately, and while applicable in principle, the instantiation of the generalized strategic hinge in our setting turns out to be intractable, as it results in a doubly-nested objective with non-convex (and even discontinuous) constraints. As an alternative, our approach builds on Eq. (11) and modifies it to support ambiguity.

### 5.1. The ambiguous strategic hinge

Our proposed loss function relies on the fact that while ambiguity affects user behavior, the classifier itself remains linear. This allows us to maintain the general form of Eq. (11) and express ambiguity only through how margins are measured. We begin with the standard notion of strategic margins, and then show how to extend this to account for ambiguity.

**Strategic margins.** One interpretation of Eq. (11) is that, relative to the non-strategic hinge, each point $x$ is provided an additional 'reward' of $r = (2/\alpha)\|w\|_2$ score points, given by the maximal distance any point can move $(2/\alpha)$ converted to margin units ($\|w\|_2$). This induces an 'effective' decision boundary, namely the hyperplane $w^\top x + b + r = 0$, which partitions the raw input space $\mathcal{X}$ into points that *will* receive positive vs. negative predictions *after* (possible) strategic movement. Margins are then measured relative to this effective decision boundary, and $h$ is penalized in the loss accordingly. Note Eq. (11) differs from the conventional non-strategic hinge only in this reward term, which fully accounts for strategic behavior.

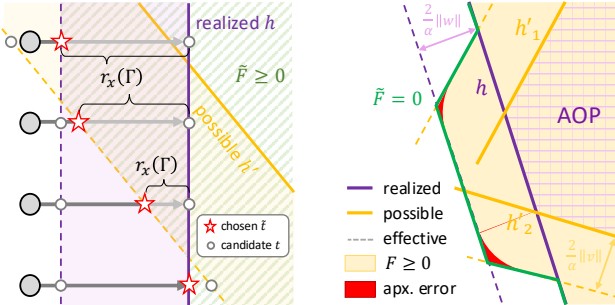

*Figure 2.* **Left:** The ambiguous strategic hinge loss (Eq. (12)) reduces prediction penalties by administering 'rewards' $r_x(\Gamma)$ based on directional distances to the effective decision boundary ($\widetilde{F} = 0$). **Right:** An outer polyhedral approximation of the decision boundary, enabling a closed-form solution at often minimal distortion.

**Ambiguity-aware margins.** Maintaining a similar perspective, our approach will be to modify the reward in Eq. (11) to account for both strategic behavior *and* ambiguity. The first step is to notice that under ambiguity, the effective decision boundary may no longer be linear (see Fig. 1). This is since points that move must now reach the AoP, which in itself is non-linear. The implication is that two points $x, x'$ which are equidistant from $h$ may require a different reward if their distance to the AoP is different. Thus, the reward must now be an input-depend function of the ambiguity set, denoted $r_x(\Gamma)$. Our proposed *ambiguous strategic hinge* is:

$$\ell_{\text{amb}}(x, y; h, \Gamma) = \max\{0, 1 - y(w^\top x + b + r_x(\Gamma)\} \quad (12)$$

Note Eq. (12) differs from Eq. (11) only in the reward $r_x(\Gamma)$. Note also that $r_x(\Gamma)$ is the only component of the loss that is affected by ambiguity, here through its dependence on $\Gamma$.

The reward $r_x(\Gamma)$ should express the distance, in score units, from the point $x$ to the effective decision boundary, which now depends on $\Gamma$. Let $F$ be a function characterizing this boundary as $F(x) = 0$, with $F(x) \geq 0$ determining the region of positive predictions. For now we will keep $F$ abstract, but will return to discuss its structure shortly, though generally it will be non-linear (Trachtenberg & Rosenfeld, 2025). Since points are ultimately classified by $h$, it is natural to measure margins towards it. Hence, for a given example $x$, let $z^*$ be the closest point in the direction of $h$ that is on the decision boundary defined by $F$, namely:

$$(z^*, t^*) = \underset{z \in \mathbb{R}^d, t \in \mathbb{R}}{\arg\min} \; t \; \text{ s.t. } \; z = x + tw, \; F(z) = 0 \quad (13)$$

and noting $z^* = x + t^*w$. The reward for $x$ is then given by:

$$r_x(\Gamma) = \max\{0, -w^\top(z^*) + b\} \quad (14)$$

where max ensures only points with $h(x) = 0$ are rewarded. Note that Eq. (12) recovers Eq. (11) since for the special case of $\Gamma = \{h\}$, Eq. (14) gives $r_x(\Gamma) = \frac{2}{\alpha}\|w\|_2$. This

idea is illustrated in Fig. 2 (left). For larger ambiguous sets, the reward can only decrease. Under full ambiguity $r_x = 0$, recovering the non-strategic hinge.

**Optimization.** Although $\ell_{\mathrm{amb}}$ admits a simple form, it requires solving Eq. (13), which entails two main challenges. First, since the effective decision boundary may not be linear, the constraint $F(x) = 0$ can be intractable. Second, $z^*$ is the solution to an argmin problem, and so is non-differentiable. Our solution is to replace $F$ with a simple yet tight approximation that overcomes both issues.

Consider a classifier $h$ and ambiguity set $\Gamma \ni h$. A given point $x$ will receive a positive prediction if either: (i) $h(x) = 1$ initially, hence $x$ will not need to move, or (ii) $h(x) = 0$, but the distance from $x$ to $\mathbb{P}(\Gamma)$ is at most $2/\alpha$, and so $x$ can move, and will. Since the effective positive region is the union of the above two conditions, $F$ is given by a union of a halfspace, and a convex set defined by all points at distance at most $2/\alpha$ from the AoP. This makes it difficult to work with. However, we can simplify $F$ considerably without much distortion by replacing it with its outer polyhedral approximation, denoted $\widetilde{F}$. This is obtained by shifting all $h' \in \Gamma$ by $(2/\alpha)\|v\|_2$ (to account for strategic behavior), taking their intersection (to account for ambiguity), and then taking the union with $h$. See illustration in Fig. 2 (right).

The resulting decision boundary admits an explicit form:

$$\widetilde{F}(x) = \min\{w^\top x + b, \max_{v,a \in \Gamma} v^\top x + a + \frac{2}{\alpha}\|v\|_2\} \quad (15)$$

For PD costs of the form $\phi(\|A^{1/2}(x - x')\|_p)$, the 2-norm becomes $\phi^{-1}(\frac{2}{\alpha})\|A^{-1/2}w\|_*$, where $\|\cdot\|_*$ is the dual norm. Under weak agreement, namely $w^\top v \geq 0$ for all $v \in \Gamma$, Eq. (13) with $\widetilde{F}$ in place of $F$ can be solved in closed-form:

$$\tilde{t} = \min\left\{t_{w,a}, \max_{v,a \in \Gamma}\{t_{v,b}\}\right\} \quad (16)$$

Here $t_{w,b}$ and $t_{v,a}$ are 'candidate' margins given by:

$$t_{w,b} = -\frac{w^\top x + b}{w^\top w}, \quad t_{v,a} = -\frac{v^\top x + a + \frac{2}{\alpha}\|v\|_2}{v^\top w} \quad (17)$$

where the max and min implement the union and intersection operators defining $\widetilde{F}$, as described above. Intuitively, $\tilde{t} = t_{v,a}$ if some shifted possible $h_{v,a}$ is closer to $x$ (in the direction of $w$) than the unshifted $h_{w,a}$, and otherwise $\tilde{t} = t_{w,b}$. In Fig. 2 (left), the top three points correspond to the first case, and the bottom point corresponds to the second.

The approximate reward can then be simplified to:

$$\tilde{r}_x(\Gamma) = \max\{0, -w^\top(\tilde{z}) + b\}, \quad \tilde{z} = x + \tilde{t}w \quad (18)$$

and the final ambiguous strategic loss is obtained by plugging $\tilde{r}$ from Eq. (18) into Eq. (12) as a substitute for $r$:

$$\tilde{\ell}_{\mathrm{amb}}(x, y; h, \Gamma) = \max\{0, 1 - y(w^\top x + b + \tilde{r}_x(\Gamma))\} \quad (19)$$

Since all components of $\tilde{\ell}_{\mathrm{amb}}$ are differentiable, the entire loss permits efficient gradient computation. In practice we found it useful to replace the hard max and min in Eq. (16) in the formula for $\tilde{t}$ with an expectation over softmax and softmin probabilities, respectively. This allows gradients to flow through all $h' \in \Gamma$ for every point $x$, rather than just the single $h'$ whose shifted boundary is closest. For continuous ambiguity we take the (soft)max over the set of classifiers encountered while running Algo. (1). Since all classifiers $h_{v,a}$ have shared parameterization, typically all entries of the variables $\underline{w}, \bar{w}$ are updated at each step.

**Approximation.** As Fig. 2 (right) shows, $\widetilde{F}$ approximates $F$ well expect for possible distortion near its corners. In Appendix E we analyze how this effects the quality of loss approximation. Intuitively, the loss is exact, namely $\tilde{\ell}_{\mathrm{amb}} = \ell_{\mathrm{amb}}$, whenever the directional projection falls on a face of $F$. When it falls on a 'rounded' corner, the approximation depends on the angle of the corresponding intersecting classifiers.

# 6. Experiments

In this section we demonstrate our approach for learning under ambiguity. We begin with synthetic data, and then proceed to a semi-synthetic setting using real data and simulated user behavior. For full experimental details see Appendix B. All code is publicly available at https://github.com/BML-Technion/AmbSC.

## 6.1. Synthetic data

**Separable case.** We begin with a simple setup to demonstrate how ambiguity can be leveraged for improving learning outcomes. We focus on discrete ambiguity, and generate data according to Fig.1 (B). Here points $x$ are sampled uniformly from $[-1, 1]^2$ and labels are $y = \mathbb{1}\{\min\{f_0(x), \max f_1(x), f_2(x)\} \geq 0\}$ for linear $f_i$ as in the illustration. To ensure a positive margin we discard points near the decision boundary. Training with a naïve (i.e., non-strategic) loss recovers $f_0$, which attains 62.9% strategic accuracy. Training with the standard strategic hinge loss similarly reaches 88.4% strategic accuracy, this by learning a shifted $f_0$. This is equivalent to our approach with $k = 1$. However, with $k = 2$, accuracy improves to 94.5%, and with $k = 3$ it reaches 99.2%. Here learning recovers $f_0$ as the classifier, and adds shifted $f_1, f_2$ to the ambiguity set. This is illustrated in Fig. 4 in Appendix C.1. As for social outcomes, notice that the non-strategic classifier errs on negative inputs, giving rise to gaming behavior. Contrarily, the standard strategic classifier errs only on positive inputs, which results in high social burden. This demonstrates how two classifiers with matching (subtopimal) strategic accuracy can induce very different social outcomes. Our approach with $k = 3$ mitigates both types of negative out-

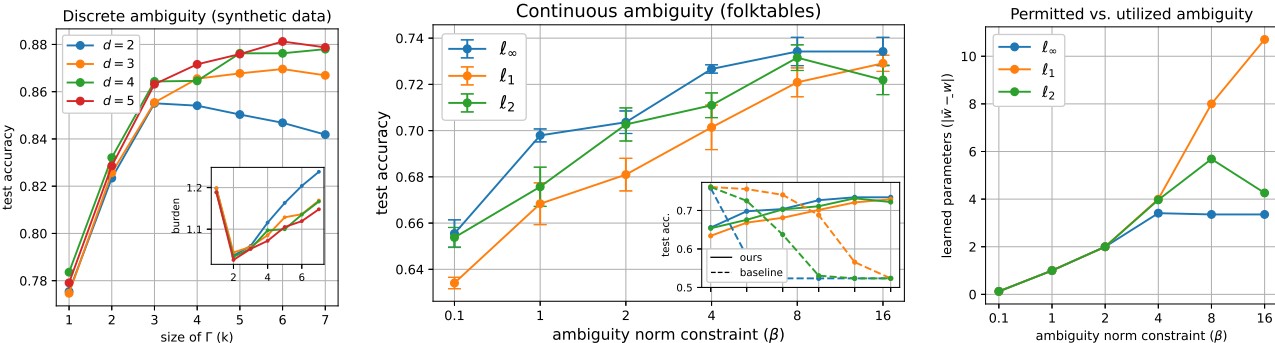

*Figure 3.* Accuracy of our approach generally increases with the degree of permitted ambiguity, both in the discrete case with $k$ **(left)** and in the continuous case with $\beta$ **(center)**. Optimal accuracy is typically attained at the correct, rather than maximal, amount of ambiguity **(right)**.

comes, and achieves maximal accuracy with less social burden or gaming behavior. Notice that the strategic baseline portrays a no ambiguity setting, whereas the non-strategic baselines (on non-strategic data) portrays full ambiguity. Results show that neither extreme is desirable.

**Gaussian data.** Next we consider the classic example of class-conditional multivariate Guassian data, $p(x|y) = \mathcal{N}(\boldsymbol{\mu}_y, \boldsymbol{\Sigma}_y)$. Given $d$, we set $\mu_+ = (1, 0, ..., 0)$, $\Sigma_+ = \text{diag}(0.5, 0.3, ..., 0.3)$, $\mu_- = (-0.5, 0, ..., 0)$, $\Sigma_- = \text{diag}(2, 8, 2, ..., 2)$ $p(y) = 1/2$. Note the larger variance for $p(x|y = 0)$ is designed to amplify the potential effect of ambiguity. Fig. 3 (left) shows strategic accuracy for increasing $k$ and varying $d$. Here the standard strategic baseline corresponds to $k = 1$. As $k$ grows, our approach exhibits increasing accuracy gains, suggesting it utilizes its capacity for ambiguity effectively. The inset graph shows that social burden first decreases with $k$, and then moderately increases. This shows that the social optimum is attained at some mid-degree of ambiguity, for which regulation can aim.

### 6.2. Real data

We use the `folktables` dataset (Ding et al., 2021) and in particular the `ACSEmployment` task. Here we focus on continuous ambiguity, and examine the effects of regulation by varying its degree and type. This is achieved by adding to the objective a hard constraint on ambiguity of the form $\|\bar{w} - \underline{w}\|_q \le \beta$ for $q \in \{1, 2, \infty\}$, where larger $\beta$ permits more ambiguity (or less transparency) for learned models. Here we compare our approach to a baseline that first learns using the standard strategic hinge, and then sets $\underline{w}, \bar{w}$ by taking the largest intervals around $w$ satisfying the norm constraint. That is, the baseline trains a strategic model assuming full transparency, and then adds ambiguity post-hoc.

Fig. 3 (center) shows accuracy for increasing values of $\beta$ and across norm types $q$. When $\beta$ is very low, our approach does not improve over the post-hoc baseline, which is tailored for transparency ($\beta = 0$; see inset). This

is likely due to the increased complexity of our objective, which is harder to optimize. However, as $\beta$ increases and more ambiguity is allowed, our approach utilizes this freedom to learn increasingly accurate classifiers and matching ambiguity sets. Meanwhile, baseline performance deteriorates and quickly reaches 50% accuracy, and is highly sensitive to norm type. This highlights a potential tradeoff for the system between accuracy and information control, which applies whenever (partial) opacity is desired or required. Note how for our method accuracy plateaus at interim $\beta$ values (which differ per norm). Fig. 3 (right) shows that at these points the effective $\|\bar{w} - \underline{w}\|$ of the learned model plateaus, meaning that the constraint w.r.t. $\beta$ is loose. Together, these again suggest that a 'correct' amount of ambiguity is optimal. Social burden is generally much lower for our method, but its dependence on $\beta$ is more nuanced; see detailed analysis in Appendix C.2.

## 7. Discussion

Most decisions inherently involve uncertainty; this should hold also for decisions made by users in response to learned models. Our work studies ambiguity as a possible form of such uncertainty, one that is likely to arise when systems have control over what information is released. Our results suggest that, perhaps surprisingly, full opacity is not always in the system's best interest. This motivates a hybrid approach to regulating transparency that sets limits on the degree of permitted ambiguity, but permits flexibility within this scope. One question is whether this idea can be applied to settings where learned models are complex, and systems seek to share simpler or interpretable proxy models. Another question is whether competition between systems over users in a market enhances ambiguity, or rather, encourages transparency. We leave these questions and others to future work.

## Acknowledgements

The authors would like to thank Inbal Talgam-Cohen for her dilligent advice and insightful feedback. This work is supported by the Israel Science Foundation grant no. 278/22.

## Impact statement

Our work aims to study the effect of ambiguity on learning and social outcomes through its impact on user behavior. This is motivated by the need to better understand the role and importance of transparency in strategic learning settings. Transparency in machine learning is typically considered as a normative requirement. Instead, we seek to understand whether and when transparency will emerge organically through strategic interactions, driven by endogenous incentives, and in a way that benefits all parties involved. Towards this, our approach extends the conventional framework of strategic classification to account for ambiguity. Our revised framework allows the learner control over ambiguity, but within the constraints imposed exogenously by a regulator. This highlights the possible interaction that can arise between systems, their users, and the broader environments in which they operate.

At the same time, it is important to remember that our results rely on the standard assumptions of strategic classification, which captures the tension that can arise in such settings, albeit in a simplified and even stylized manner. Using our proposed learning approach in practice must be done with care, in consideration of the assumptions it was developed under, and with responsibility regarding its potential social impact once deployed. Similarly, our conclusions regarding regulation should be interpreted within this scope. Our hope is that our work stimulates discussion regarding the benefits of emergent transparency, the need for balancing between a system private information and its users' access to it, the possibilities of aligning incentives via bounded ambiguity, and the role of machine learning in this process.

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

# A. Proofs

## A.1. General lemmas

**Proof of Lemma 1.** In the offset ambiguity case, the strategic response $\Delta$ is defined as:

$$\Delta_{w,b}^{\Gamma}(x) = \Delta_{w,b}^{\{\underline{b},\bar{b}\}}(x) = \operatorname*{argmax}_{x'} \min_{b \in [\underline{b},\bar{b}]} h_{w,b}(x') - \alpha c(x, x') \tag{20}$$

In this case, notice that for all $x$ the inner min is always achieved by $b = \underline{b}$, as this always induces the classifier that is furthest in the AoP (from any $x$). This means that Eq. (20) can be rewritten as:

$$\Delta_{w,b}^{\Gamma}(x) = \operatorname*{argmax}_{x'} h_{w,\underline{b}}(x') - \alpha c(x, x') \tag{21}$$

which recovers the standard strategic response under $h_{w,\underline{b}}$ and $\alpha$. To show equivalence, we next show that exists a cost scale $\alpha_\Gamma$ for which the following holds:

$$h_{w,b}(\Delta_{w,\underline{b}}(x; \alpha)) = h_{w,b}(\Delta_{w,b}(x; \alpha_\Gamma)) \tag{22}$$

I.e., that classifying users using $h_{w,b}$ under ambiguity (and cost scale $\alpha$) is equivalent to using $h_{w,b}$ with no ambiguity (and under cost scale $\alpha_\Gamma$). By Eq. (21), $\alpha_\Gamma$ can be derived by comparing the distance between the decision boundaries of two effective classifiers: $w^\top x + b + \frac{2}{\alpha_\Gamma}\|w\|_2$, and the shifted $h_{w,\underline{b}}(x) = w^\top x + \underline{b} + \frac{2}{\alpha}\|w\|_2$. Equating these gives:

$$\frac{2}{\alpha_\Gamma}\|w\|_2 + (b - \underline{b}) = \frac{2}{\alpha}\|w\|_2$$

$$\alpha_\Gamma = \frac{1}{1 - \frac{b - \underline{b}}{2\|w\|_2}\alpha}\alpha$$

Note that $\alpha \leq \alpha_\Gamma$, and is upper-bounded by:

$$\alpha_\Gamma \leq \alpha\left(1 - \frac{\bar{b} - \underline{b}}{2\|w\|_2}\alpha\right)^{-1}$$

For cost of type $\phi(\|A^{1/2}(x - x')\|_p$ we need to modify the term $\frac{2}{\alpha}\|w\|$ with the term $u^*\|A^{-1/2}w\|_*$ where $u^*$ satisfies that its the highest $u$ such that $\phi(u^*) \leq \frac{2}{\alpha}$ and $\|\cdot\|_*$ is the dual norm of $\|\cdot\|_p$ (as presented in (Rosenfeld & Rosenfeld, 2024)). $\qquad\square$

**Proof of Lemma 2.** Let $\mathcal{S}_{cont}$ be the set of feasible solutions of Eq. (7), and let $\mathcal{S}_{disc}$ be the set of feasible solutions of Eq. (9), namely:

$$\mathcal{S}_{cont} = \{x' : w^\top x' + b \geq 0, \ \forall w \in [\underline{w}, \bar{w}]\}$$

$$\mathcal{S}_{disc} = \{x' : w^\top x' + b \geq 0, \ \forall w \in \Omega\}$$

We show that $\mathcal{S}_{cont} = \mathcal{S}_{disc}$.

$\mathcal{S}_{cont} \subseteq \mathcal{S}_{disc}$: Since $\Omega \subseteq [\underline{w}, \bar{w}]$, every solution $x'$ in $\mathcal{S}_{cont}$ satisfies $\forall w \in \Omega$, $w^\top x' + b \geq 0$. Therefore, $x'$ is also a solution to $\mathcal{S}_{disc}$.

$\mathcal{S}_{disc} \subseteq \mathcal{S}_{cont}$: Let $x' \in \mathcal{S}_{disc}$. By definition, we know that $\forall w \in \Omega$, $w^\top x' + b \geq 0$, or essentially $\min_{w \in \Omega} w^\top x' + b \geq 0$. Suppose by contradiction that $x' \notin \mathcal{S}_{cont}$. That is, there exists $w \in [\bar{w}, \underline{w}]$ such that $w^\top x' + b < 0$. Based on this $w$, we will construct $w'$ which yields even lower score than $w$, such that $w' \in \Omega$. Recall that $w^\top x' = \Sigma_i w_i x_i'$. Then for each $i$:

1. If $x_i' < 0$, pick $w_i' = \bar{w}_i$. It holds that $w_i' \cdot x_i' \leq w_i \cdot x_i'$, since $w_i' \geq w_i$.

2. If $x_i' \geq 0$, pick $w_i' = \underline{w}_i$. It holds that $w_i' \cdot x_i' \leq w_i \cdot x_i'$, since $w_i' \leq w_i$.

Therefore, we get that $w'^\top x' + b < 0$ also. This is a contradiction, since $w' \in \Omega$ and hence $x' \notin \mathcal{S}_{disc}$.

Eq. (7) and Eq. (9) minimize the same objective. Hence, from the equivalence of their feasibility sets we get that they share the same optimal solutions. $\qquad\square$

***Proof of Lemma 4***. We prove both directions:

- **If the AoP is not empty:** If $b \geq 0$, then we finish. Otherwise, $b < 0$. Since the AoP is not empty, there exists a point $x$, such that $\forall w \in [\underline{w}, \bar{w}]$ it holds that $w^\top x + b \geq 0$. Note that $x$ is not the 0 vector since $b < 0$.

  Assume by contradiction that $\underline{w}, \bar{w}$ do not agree on any coordinate. That is, and since $\underline{w} \leq \bar{w}$, it holds that $\underline{w}_i < 0, \forall i$ and $\bar{w}_j \geq 0, \forall j$. Now, we will construct a vector $\tilde{w} \in [\underline{w}, \bar{w}]$ in the following way: For each $i \in [d]$, if $x_i \geq 0$, then $\tilde{w}_i = \underline{w}_i$; else, $\tilde{w}_i = \bar{w}_i$. Since $b < 0$, and there exists a coordinate $i$ such that $x_i \neq 0$, it holds that $\tilde{w}^\top x + b < 0$. This is a contradiction since $x$ is in the AoP.

- **If $b \geq 0$ or $\underline{w}_i \cdot \bar{w}_i > 0$ for some $i \in [d]$:** First if $b \geq 0$, a trivial solution is $x_i = 0, \forall i \in [d]$. Therefore, for each $w$ in $[\underline{w}, \bar{w}]$, we get $w^\top x + b = b \geq 0$. Otherwise, we have $b < 0$. Denote by $i$ the coordinate that $\underline{w}_i \cdot \bar{w}_i > 0$. This means that both $\underline{w}_i, \bar{w}_i$ have the same sign. We now show that there exist $x$ such that $x$ is in the AoP. Assume w.l.o.g that $0 \leq \underline{w}_i \leq \bar{w}_i$. We can take $x_j = 0, \forall j \neq i$ and in the $i^{\text{th}}$ coordinate we take $x_i$ such that $\underline{w}_i \cdot x_i \geq -b$. For every $w \in [\underline{w}, \bar{w}]$ and for $b$, it holds that $w^\top x + b \geq \underline{w}_i \cdot x_i + b \geq 0$. Therefore, the AoP is not empty.

$\square$

## A.2. Separating oracle

***Proof of Lemma 3***. The construction of $w^*$ is in the same way as in the proof of Lemma 2. Let $x$ be a non-feasible solution for $\Omega$. We will construct the "worst" classifier $w^* \in \Omega$ that yields the lowest score $w^{*\top} x + b$:

1. If $x_i < 0$, pick $w^{*\prime}_i = \bar{w}_i$. It holds that $w^{*\prime}_i \cdot x_i < 0$ and is smaller than $\underline{w}_i \cdot x_i$.

2. If $x_i < 0$, pick $w^{*\prime}_i = \underline{w}_i$. It holds that $w^{*\prime}_i \cdot x_i < 0$ and is smaller than $\bar{w}_i \cdot x_i$.

Therefore, $w^{*\top} x + b = \operatorname{argmin}_{w \in \Omega} w^\top x + b$, and since $x$ is non feasible, $w^{*\top} x + b < 0$. Hence, $w^*$ separates $x$ from the feasible set defined by $\Omega$ and $b$. $\square$

## A.3. Projection algorithm (Algo. 1)

***Proof of Theorem 1***. First, if by the end of the algorithm $\mathcal{C} = \Omega$ then correctness is trivial. We therefore focus on the case where the algorithm terminates early and the returned $x^*$ is computed with $|\mathcal{C}| \subset \Omega$ that satisfies the condition in line 5.

**Correctness:** Given $x'$, let $w' \in \Omega$ be the classifier that yields the worst score on $x'$. As per Lemma 3, all other classifiers yield higher scores than $w'$. Thus, if $w'^\top x' + b \geq 0$, then $\forall w \in [\underline{w}, \bar{w}]$, $w^\top x' + b \geq 0$, meaning that $x' \in \mathbb{P}$.

**Optimality:** Let $\mathcal{W}$ denote the set of extreme classifiers induced by $\underline{w}$ and $\bar{w}$.

Suppose, for contradiction, that there exists $\tilde{x}$ such that $c(x, x') > c(x, \tilde{x})$ and $w^\top \tilde{x} + b \geq 0$ for all $w \in \mathcal{W}$.

Let $\mathcal{C}$ be the set of constraints used by Algorithm 1 to produce $x'$. We know that $\mathcal{C} \subseteq \mathcal{W}$, so $\tilde{x}$ satisfies all constraints in $\mathcal{C}$.

Therefore, in the iteration of the algorithm with the constraint set $\mathcal{C}$, both $x'$ and $\tilde{x}$ are feasible solutions. the chosen solution is the one which minimizes the distance to $x$, and since $x'$ was returned, it holds that $c(x, x') \leq c(x, \tilde{x})$. This is a contradiction. $\square$

# B. Experimental details

## B.1. Synthetic Data

### B.1.1. SEPARABLE CASE

Data is generated as follows:

- Inputs $x = (x_1, x_2)$ are sampled uniformly from a 2D rectangle, where $x_1 \in [-6, 4]$, $x_2 \in [-10, 10]$.

- Labels are assigned positive iff $x$ satisfies either:

- $x_1 \geq 1$, or
- $x_1 + x_2 + 2 \geq -2$, $x_1 - 1 \geq -2$ and $x_1 - x_2 + 2 \geq -2$

- Point near the decision boundary are removed to create a margin.

In each trial we sample 1000 examples, and randomly split the data 50:10:40 into train, validation, and test sets.

Training for all methods was done using gradient descent with 500 epochs, learning rate 0.006, and weight decay 0.001. For the non-strategic baseline we used the standard hinge loss. For $k = 1$ we used the standard strategic hinge loss (Eq. (11)). For $k > 1$ we used our proposed ambiguous strategic hinge (Eq. (12)), with small penalty term (coefficient 0.001) that enforces agreement between the realized and all possible classifiers (see B.1.3).

### B.1.2. GAUSSIAN DATA

We experimented with increasing dimensions $d \in [2, 3, 4, 5]$ and ambiguity set size (in this case, the number of possible classifiers) $k = [1, 2, 3, 4, 5, 6, 7]$. Note that $k = 1$ recovers the standard strategic classification setting. Given $d$, data is sampled from class-conditional multivariate Gaussians with the following parameters:

1. For the positive class, $\mu_+ = [1, 0, 0, ..., 0]$ and $\Sigma_+ = \mathrm{diag}(0.5, 0.3, ..., 0.3)$

2. For the negative class, $\mu_- = [-0.5, 0, 0, ..., 0]$ and $\Sigma_- = \mathrm{diag}(2, 8, 2, 2, ..., 2)$

The label prior was set to $P(y = 1) = p(y = -1) = 0.5$, and the cost scale in $\Delta$ was set to $\alpha = 1$. Results were averaged over twenty random instances of the data.

### B.1.3. OPTIMIZATION

**Parameter initialization.** In all experiments we initialized the weights of the true classifier $w$ by sampling each entry from a standard Gaussian distribution, and then normalizing as $\|w\|_2 = 1$. All other classifiers $v \in \Gamma$ were initialized by taking $w$, adding Gaussian noise with small variance 0.8 per entry, and normalizing. All offset terms were initialized to $b = -1$.

**Regularization.** In addition to $\ell_2$ regularization, we add to the objective a term that penalizes the model for disagreement between the true $w$ and any of the possible $v$, measured as $\cos(w, v)$.

**Optimization.** We used Adam with stochastic mini batches. For the separable data experiment, we trained for 500 epochs, with learning rate 0.006, weight decay 0.001, and agreement regularization coefficient 0.001. The temperature parameter for the softmax and softmin of the ambiguous strategic hinge loss was set to 0.15. For the Gaussian experiment, we used 750 epochs, learning rate 0.006, weight decay 0.02, agreement regularization 0.08 and temperature 0.25.

## B.2. Real Data

### B.2.1. FOLKTABLES

**Data description.** For our main experiment we use a subset of the `folktables` dataset, which is publicly available at https://github.com/socialfoundations/folktables.git. We focus on the employment prediction task, where the goal is to predict whether an individual is employed or not. We considered data points from Alabama ('AL'), which contains 47,777 examples. To obtain a balanced dataset in terms of labels, each instance of our experiment uses a random subset of 9,000 examples for which $|\{y = 1\}| \approx |\{y = -1\}|$.

**Features and labels.** We used the following features: 'AGEP'- age (integer), 'SCHL'- educational attainment (ordinal from 1 to 24), 'MAR'- marital status (ordinal from 1 to 5), 'RELP'- religious affiliation (ordinal from 1 to 17), 'ESP'- employment status of parents (ordinal from 1 to 8), 'CIT'- citizenship status (ordinal from 1 to 5) and 'MIL'- Military Service (ordinal from 0 to 4). All features where scaled to $[0, 1]$. Labeled were set by binarizing the attribute 'ESR' - employment status.

**Data generation.** For each experimental instance we sampled 9,000 random class-balanced example. These were split 40:30:30 into train, validation, and test sets. Results were averaged over five random instances. For strategic behavior, we set the cost scale to $\alpha = 6$ in all instances, chosen so that roughly between 10-40% moved across all conditions.

**Learning algorithms.** Our approach uses the ambiguous strategic hinge loss (Eq.(12)). For the standard (non-ambiguous) strategic classification we used the standard strategic hinge (Eq.(11)). For all methods we used $\ell_2$ regularization.

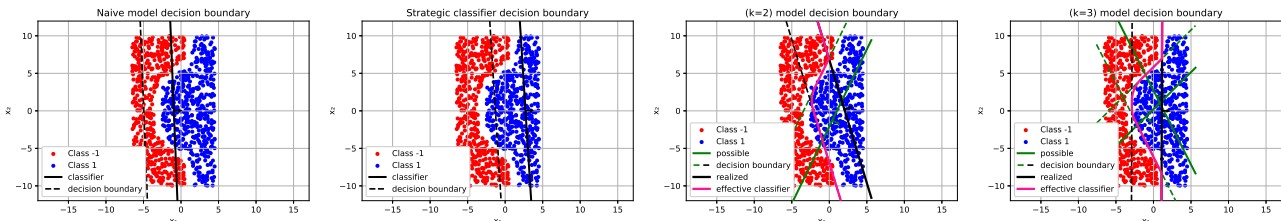

*Figure 4.* Learned models for the separable data experiment. From left to right: naïve (non-strategic), standard strategic ($k = 1$), ambiguous strategic with $k = 2$, and ambiguous strategic with $k = 3$. For $k = 2, 3$ we also plot the effective classifier that is created based on the ambiguity set.

**Hyperparameters.** All methods use similar hyperparameters and the same training procedure: 400 epochs, batch size 128, learning rate 0.001, and $\ell_2$ regularization coefficient 0.005. Temperature for the softmax and softmin was set to 0.2.

**Initialization.** To initialize $\underline{w}, \bar{w}$ we sample a random vector $v$ of unit norm, created a $d$ dimensional box which centers in $v$, and set $\underline{w}$ by taking the minimum coordinates of the box and $\bar{w}$ by taking the maximum coordinates of the box. We then set $w$ such that each $w_i$ was sampled uniformly in the range $[\underline{w}_i, \bar{w}_i]$. The offset term was initialized as $b = -0.1$.

### B.3. Training and optimization

**Implementation.** All code was implemented in Python and using PyTorch.

**Optimization.** For optimization, we use ADAM and split the data to mini batches. After each gradient step we applied a projection step to enforce the parameter constraint $w \in [\underline{w}, \bar{w}]$ and the continuous ambiguity norm constraint $\|\bar{w} - \underline{w}\|_q \le \beta$. Projections (if needed) were computed by solving an appropriate convex program using CVXPY. The complexity of the ambiguous strategic loss in this case makes the objective mildly sensitive to initialization. For this, we ran 10 different random initializations per instance, and chose the model giving the highest accuracy on the held-out validation set.

**Regularization.** Continuous ambiguity requires two additional (hard) constraints, implemented as (soft) penalties in the objective. The first constraint ensures that all possible $h' \in \Gamma$ agree with the true $h$. This is conceptually similar to the discrete case, but cannot be implemented via enumeration. However, a sufficient condition is to calculate $\min_{v \in \Gamma} v^\top w$, and penalize by it. The argmin is simply the point-wise argmin over entries $\underline{w}_i, \bar{w}_i$. The second constraint ensures that the AoP in each step is not empty. For this, we use Lemma 4 and penalized based on $\max_i \underline{w}_i \cdot \bar{w}_i$. For both constraints we use an exponentially decaying penalty.

## C. Additional experimental results

### C.1. Separable data

Figure 4 graphically illustrates data and learned model for each condition of our experiment. The naïve model (left) assumes no strategic behavior and is therefore highly susceptible to gaming behavior, and reaches 62.9% accuracy. However, it entails very low burden. The standard strategic model ($k = 1$; center-left) assumes full transparency. This helps mitigate gaming, and achieves 88.4% accuracy. This however results in many false positives, which induces very high social burden. An ambiguous strategic classifier with $k = 2$ (center-right) improves to 93.7% accuracy – showing that even mild ambiguity can greatly reduce errors, here by almost half. Finally, the ambiguous strategic classifier with $k = 3$ (right) achieves an almost optimal 97.7% accuracy. Notice how the realized classifier is in fact the same as the naïve classifier, while the possible classifiers help define an effective AoP. This model does not only give high accuracy, but also mitigates gaming and has very low social burden.

### C.2. Real data – social burden analysis

Figure 5 plots the relation between accuracy and social burden for learned models corresponding to all experimental conditions and initializations. Results show two phenomena. First, albeit noisy, the general trend is that social burden is highest at interim accuracy levels. For high-accuracy models it is consistently lower, whereas low-accuracy models exhibit two modes of burden, whose average is comparatively low, but its maximal levels are high in absolute terms. Second, per accuracy level, there is significant variation in social burden. This suggests that models attaining similar accuracy can induce

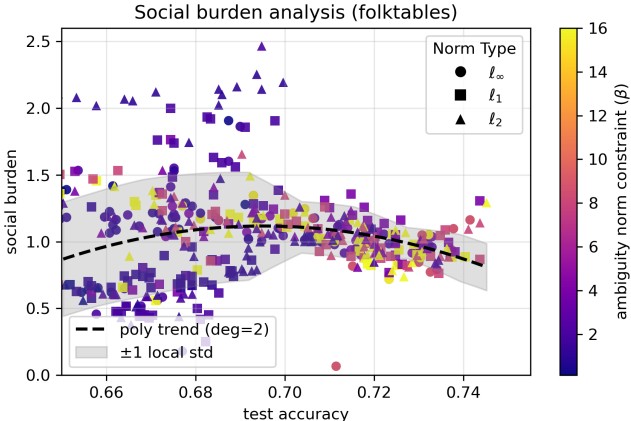

*Figure 5.* Social burden as a function of the accuracy. We can see that the burden acts in a poly trend with degree = 2, where for higher $\beta$ the points are more centered around this poly.

very different social burden levels. However, this variation is lower for high-accuracy models.

## D. Runtime analysis of projection algorithm

Although our cutting plane method (Algo. 1) does not provide efficient runtime guarantees, in practice we observed that the number of calls to the convex solver was very small, both in absolute terms and relative to $d$. We examined this in two settings: a designated synthetic setup, and our main experiment from the paper.

### D.1. Synthetic data

To evaluate the runtime of our algorithm in practice we ran a series of experiments using synthetic randomly generated data. For a given dimension $d$ and distribution, we sample $x, \underline{w}, \bar{w}$, and $b$ (ensuring $\underline{w} \leq \bar{w}$); compute the projection $x^*$ using Algo 1; and count the number of iterations required to terminate. We repeat this process 10000 times for each experimental condition, and compute statistics.

Figure 6 shows results for $d \in 3, \ldots, 9$ and three distributions: uniform in $[0, 1]$, normal, and $\mathrm{Beta}(2, 5)$. As can be seen, the maximal number of iterations per instance is $\sim 4$ on average, and 12 in the worst case. Note the average behavior is constant (or even decreasing), whereas the maximum increases only slightly with $d$. Across all instances, we did not encounter any case where the number of iterations was larger than $2d$, and the distribution is generally skewed towards lower values.

### D.2. Real data

Fig. 7 reports similar statistics but for projections computed while running our main experiment on the `folktables` dataset. Results show both average and maximum number of iterations; both are bonded by a small constant across all experimental conditions.

## E. Loss function approximation analysis

In this section we focus on analyzing both theoretically and experimentally our approximation to the loss function $\tilde{\ell}_{\mathrm{amb}}$ (Eq. (19)) compared to the real loss function $\ell_{\mathrm{amb}}$ (Eq. (12)). We use $\ell_2$ norm as the cost function throughout this analysis. Moreover, for simplicity, we assume that all classifiers in the ambiguity set share the same norm.

### E.1. Theoretical Insights

Given $h$ and $\Gamma \ni h$, the positive region can be defined as $A \cup B$, where $A$ is the positive region of $h$ and $B$ is the Minkowsky sum of $AoP(\Gamma)$ and an $\ell_2$ ball of radius $\frac{2}{\alpha}$. Since the AoP is a polytope, $B$ is an expanded, "rounded" polytope with ball segments instead of vertices. The approximation $\widetilde{F}$ outer-bounds these "corner" ball segments by extending the faces of the

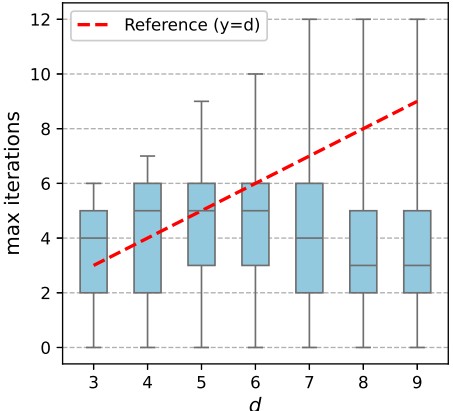
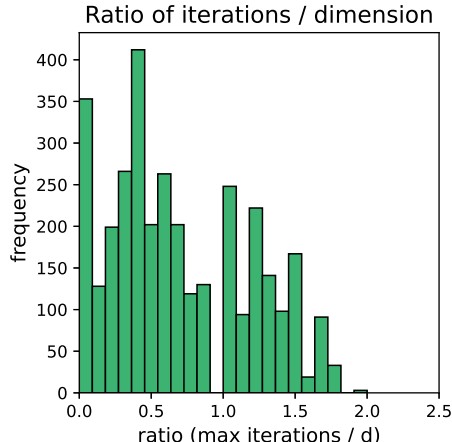

*(a)* Distribution of max iterations per dimension. Note that even for larger dimensions, the average number of iterations is still small (around 4). The outliers do not exceed $2d$ iterations for every dimension $d$.

*(b)* Histogram of the ratio between the number of iterations and the dimension $d$. Note that the bulk of the mass is below 1.0, meaning that the average number of iterations of the algorithm is at most $d$ for every $d$.

*Figure 6.* Distribution of max iterations per dimension and histogram of the ratio between the max iteration and the dimension.

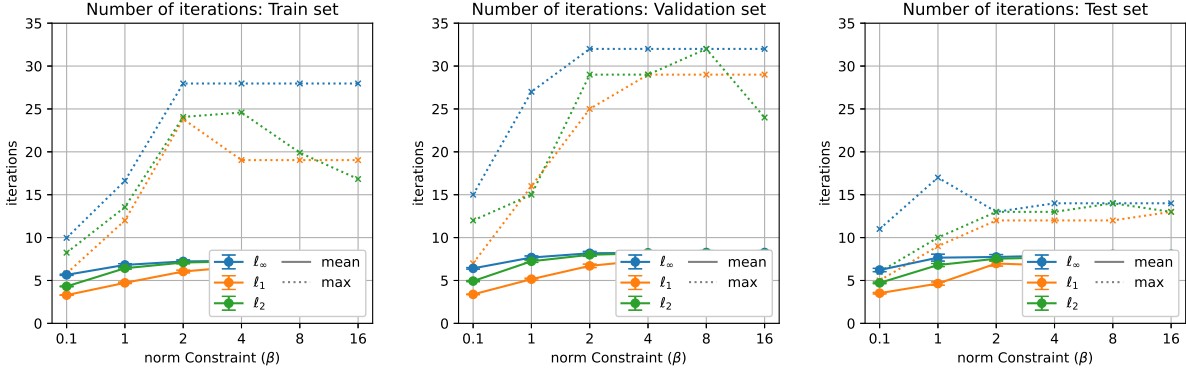

*Figure 7.* Number of iterations in folktables dataset. Note that even though we used $d = 7$ in our data, the average running time in both train, validation and test is around 7. Moreover, the maximal number of iterations is 32. This means that the algorithm works with an averaged complexity of $O(d)$.

.

(rounded) polytope to obtain a standard polytope. Note our assumption on classifier agreement implies ball segments are restricted to an orthant. We refer to Figure 2 (right) for illustration.

Denote the true and approximate ambiguous losses by $L^*$ and $\widetilde{L}$. Define the distortion as $R = |\widetilde{L} - L^*|$. We begin with some observations, and then provide a formula for $R$.

Consider a point $x$, and denote by $x'$ its projection onto $F$ in the direction of $w$. First, notice that $\widetilde{F}$ and $F$ agree on all the faces of $\widetilde{F}$. This implies the following observation:

**Observation 2.** $\widetilde{L}$ *might be approximate only if $x'$ lies on a ball segment.*

Conversely, if $x'$ lies on a face of $B$ or on $A$, then $\widetilde{L}$ is exact and $R = 0$. We expect this to be the common case, as in our experiments below.

We next provide an upper bound on $R$ in the 2-dimensional case, as it sets the basis for understanding the approximation of our loss function.

**Lemma 5.** *Let $x$ with $x' = x + t^* w$ on a ball segment $C$ whose center $c$ is a vertex of the AoP. Define by $\beta$ the angle*

*between the line from $c$ to $x'$ and the normal of $-w$. Then:*

$$|\widetilde{L} - L^*| \leq \frac{2}{\alpha}\|w\|(1 - \cos(\beta))$$

*and*

$$\beta \leq \max_{h_1,h_2 \in H(c)} 180° - angle(h_1, h_2)$$

*Where $H(c)$ includes all classifiers from $\Gamma$ intersecting at $c$.*

*Proof.* Since $x, y, w$ are the same for $L^*$ and $\widetilde{L}$, it suffices to bound the difference, measured in margin units, between the reward term of $L^*$, denoted by $r_x^*(\Gamma)$ and the reward term of $\widetilde{L}$ denoted by $\tilde{r}_x(\Gamma)$. This will be an upper bound for the difference in the loss function (since we use hinge loss, and the reward term may cause both $L^*$ and $\widetilde{L}$ to be 0).

Let $M$ be the point on the line segment from $c$ to $x'$ that intersects the hyperplane $w^T x + b + \frac{2}{\alpha}\|w\|$. Let $D$ be the orthogonal projection of $C$ into that hyperplane. Finally, let $E$ be the intersection between this hyperplane and the line passing through $x'$ in the direction $-w$. An illustration is shown in Figure 8.

Note that $|\widetilde{L} - L^*| \leq |r_x^*(\Gamma) - \tilde{r}_x(\Gamma)| = |x'E|$ in margin units.

Now using geometric claims:

- $\cos\beta = \frac{|Dc|}{|Mc|}$. $|Dc| = \frac{2}{\alpha}\|w\|$ which leads to $|Mc| = \frac{\frac{2}{\alpha}\|w\|}{\cos\beta}$.

- Since $|x'c| = \frac{2}{\alpha}\|w\|$, we have $|x'M| = \frac{2}{\alpha}\|w\|(\frac{1}{\cos\beta} - 1)$

- Finally, by the definition of $\beta$, $\cos\beta = \frac{|x'E|}{|x'M|}$.

This leads to

$$|x'E| = \frac{2}{\alpha}\|w\|(1 - \cos\beta)$$

Moreover, let $\theta_i$ denote the angle between each pair of classifiers that intersect at $c$. By the geometry of the construction, for every such angle $\theta_i$, $180° + \beta + \theta_i \leq 360°$. Equivalently

$$\beta \leq \max_{h_1,h_2 \in H(c)} 180 - angle(h_1, h_2)$$

And this completes the proof.

$\square$

The following observation follows directly from the proof:

**Observation 3.** *The bound is tight when the true classifier $h$ belongs to $H(c)$, and determines $\tilde{t}$.*

If $h$ is not in $H(c)$, then $c$ lies in the positive region of $h$ since $c$ is a vertex of the AoP. Therefore, $c$ lies on the corresponding face of $\widetilde{F}$, so $R = 0$. When $h$ determines $\tilde{t}$, the angle $\beta$ is taken with respect to it, thus the proof is tight.

Another observation is that the worst-case bound matches the distortion of the standard (non-ambiguous) strategic hinge. That is, **the worst-case ambiguous approximation is no worse than the alternative baseline**. This occurs only if both conditions hold:

- $c$ is an orthogonal vertex.

- $x'$ lies near the boundary of $C$ (where it connects to a face).

We expect this to be a rare event; in our experiments below, the average angle was 49.9 degrees.

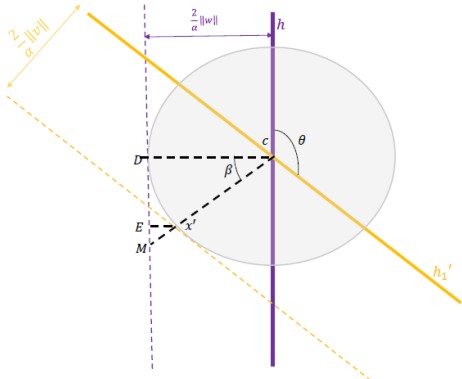

*Figure 8.* Illustration of the construction used in Lemma 5

## E.2. Experiments

For concreteness we focus on our Gaussian data experiment. Below we detail the procedure and provide a table with all results.

**Exactness.** As noted, the approximation of $x$ is exact if its directional projection $x'$ lies on a face of $B$ or on $A$. This definition is intuitive, but not easily verified. Instead, we propose an equivalent procedural definition that is computationally feasible.

Given $x$:

1. Compute $\widetilde{t}$. If it derives from the outer $min$, return EXACT.

2. Compute $x'$ and its projection to the AoP. If only one constraint is tight, return EXACT.

3. Return APX.

Note all steps above are implementable via the tools we provide in the paper.

**Empirical Distortion.** We next examine $R$ for non exact points. Note that computing the distortion requires computing $t^*$. For this, we use two properties:

1. The distance of points $z$ on the line between $x$ and $x'$ to the center $c$ of $C$ is convex in $z$.

2. $t^*$ is exactly at distance $\frac{2}{\alpha}$ from $c$.

Together, this implies we can use line search to find $t^*$. Importantly, we would like to emphasize again that the ability to compute $t^*$ (for reasonably sized sets) does not imply that learning with $t^*$ is tractable.

**Results.** We applied the above procedure to all points across all experimental conditions. For scaling we report the normalized distortion $\bar{R} = \frac{|\widetilde{L} - L^*|}{L^*}$ where $\bar{L}^*$ is the average true loss.

On average, 55% of points were exact. For non-exact points, the average $\bar{R}$ was 0.13, and the 95th percentile was 0.61. The results in detail are presented in Table (1)

*Table 1.* Summary of APX results by dimension $d$ and number of classifiers $k$ for the Gaussian experiment

| $d$ | $k$ | % APX | avg | 95% |
|-----|-----|-------|------|------|
| 2 | 2 | 13% | 0.07 | 0.18 |
| 2 | 3 | 19% | 0.07 | 0.26 |
| 2 | 4 | 29% | 0.08 | 0.33 |
| 2 | 5 | 53% | 0.11 | 0.42 |
| 2 | 6 | 59% | 0.12 | 0.52 |
| 2 | 7 | 54% | 0.11 | 0.49 |
| 3 | 2 | 7% | 0.07 | 0.36 |
| 3 | 3 | 27% | 0.08 | 0.22 |
| 3 | 4 | 45% | 0.10 | 0.44 |
| 3 | 5 | 64% | 0.16 | 0.81 |
| 3 | 6 | 67% | 0.20 | 1.00 |
| 3 | 7 | 71% | 0.25 | 1.35 |
| 4 | 2 | 19% | 0.07 | 0.21 |
| 4 | 3 | 57% | 0.09 | 0.31 |
| 4 | 4 | 68% | 0.17 | 0.91 |
| 4 | 5 | 66% | 0.19 | 1.00 |
| 4 | 6 | 73% | 0.25 | 1.34 |
| 4 | 7 | 53% | 0.18 | 0.89 |
| 5 | 2 | 7% | 0.05 | 0.20 |
| 5 | 3 | 26% | 0.07 | 0.25 |
| 5 | 4 | 32% | 0.10 | 0.38 |
| 5 | 5 | 51% | 0.13 | 0.67 |
| 5 | 6 | 60% | 0.18 | 0.98 |
| 5 | 7 | 63% | 0.19 | 1.04 |
| average | | 45% | 0.13 | 0.61 |

