# OpenReview forum: "Ambiguous Strategic Classification"
_ICML.cc/2026/Conference — ICML 2026 regular_

### Official Review · Reviewer_Amho · 2026-02-26

**Soundness:** 4
**Presentation:** 4
**Significance:** 3
**Originality:** 4
**Overall Recommendation:** 5
**Confidence:** 3

**Summary:**

The authors present ambiguous strategic classification, a novel framework that challenges the assumption that systems fully disclose their classifiers to agents applying. Their model allows for interpolation between no information and full transparency to maximize the principal’s goals. By using ambiguity, the learner can jointly optimize a true classifier and a range of potential alternatives to shape user behavior.
The authors model this as a learning task where users make decisions under ambiguity, meaning they are aware of the set of possible classifiers but have no attached probabilities of which one will be used. They assume users are risk averse and perform a maxmin best response. To solve this, the authors introduce the area of positivity as the intersection of all positive regions of the ambiguity set, and develop an algorithm to compute responses. Finally, they propose an ambiguous strategic hinge, a differentiable loss function that uses an input dependent reward term to account for nonlinear boundaries.

**Compliance With Llm Reviewing Policy:**

Affirmed.

**Final Justification:**

I will maintain my belief that this paper should be accepted. The clarifications in the rebuttal addressed the issue of uncertainty versus ambiguity in this context. Furthermore, the examples of the uncertainty structure types are useful. Therefore, I will maintain my initial score of 5.

**Key Questions For Authors:**

Can the authors better justify the ambiguity setting in real practice, and how does it compare to uncertainty performance wise?

**Limitations:**

yes

**Strengths And Weaknesses:**

Strengths:
- I think the paper is well written and clear.
- I think the explanation of linear ambiguous classifiers forming a nonlinear AOP is interesting and useful.
- Lemma 2 provides a valuable insight on how offset ambiguity compares (is equivalent to with higher cost) with standard strategic classification.

Weaknesses:
- The authors contrast their ambiguity setting with uncertainty by clarifying that there are no probabilities assigned to outcomes. I am concerned that in practice, agents will assign subjective probabilities even to ambiguous sets. A user may perceive a higher chance for one classifier and gamble on that.
- I think the paper could be improved by adding more comparison with uncertainty in strategic classification performance wise. Without this comparison, it is difficult to quantify how much of the performance gain is due to ambiguity versus general uncertainty that could be modeled with noise.

---

> ### Author Rebuttal · Authors · 2026-03-31
>
> Thank you for your positive review! We are glad you believe our paper makes a solid contribution to the strategic learning literature. Please see our response to your questions below.
>
> > The authors contrast their ambiguity setting with uncertainty by clarifying that there are no probabilities assigned to outcomes. I am concerned that in practice, agents will assign subjective probabilities even to ambiguous sets. A user may perceive a higher chance for one classifier and gamble on that.
>
> We agree this is possible, but believe this concern applies equally to any type of assumption made on user behavior. For example, assuming users respond based on subjective probabilistic beliefs risks misspecification against users that are in fact risk averse.
>
> In our mind, the main difference between ambiguity and subjective probabilistic uncertainty is that under ambiguity, the learner retains full control over the uncertainty that users face. In contrast, when users act on personal beliefs, then these constitute private information which the learner does not have access to. Anticipating user behavior in this case therefore requires either assuming such information is public (as in Cohen et al. (2024b)), or adding an information elicitation dimension to the learning framework, which makes it considerably more challenging.
>
> Overall, we view our approach as applicable to settings where the learner has reason to believe that users will be willing to invest effort in order to guarantee themselves a positive prediction (rather than hoping the dice roll in their favor).
>
>
> > I think the paper could be improved by adding more comparison with uncertainty in strategic classification performance wise. Without this comparison, it is difficult to quantify how much of the performance gain is due to ambiguity versus general uncertainty that could be modeled with noise.
>
> Thank you for this comment. While conceptually appealing, this introduces several difficulties. First, as we write above, if users have subjective beliefs, then it is unclear how the system can anticipate their behavior, let alone learn strategically; we are unaware of any practical learning algorithm that supports this form of behavior. Second, while beliefs over a discrete set of classifiers is simple to model, we do not see a direct way to parameterize a belief distribution over a continuous ambiguity set. Third, and even for discrete ambiguity, it is unclear how to compute best responses efficiently: if users have a prior $p$ over models, then utility is an expectation over binary predictions, i.e., $E_p[sign(w_{(i)}^\top x + b_{(i)})]$, which does not admit any straightforward solution, as it requires enumerating all possible combinations of correct and incorrect predictions across the set of classifiers to find the utility-maximizing $x’$.
>
>
> > Can the authors better justify the ambiguity setting in real practice, and how does it compare to uncertainty performance wise?
>
> As a motivating example, consider the task of designing a qualifying exam, e.g. for job hiring. Features represent answer correctness on questions of different topics, and weights represent the relative importance of each topic to the total score. If the firm lets candidates know the exam exactly, then candidates will prepare for the exam specifically, which will harm the firm’s ability to screen candidates effectively. As an alternative, the firm can choose to keep certain aspects of the exam ambiguous. For example, and using our proposed uncertainty structure types:
> * Offset ambiguity: The firm determines in advance the importance of each topic, but keeps the acceptance threshold ambiguous.
> * Discrete ambiguity: The firm releases a set of possible exams (e.g., each focusing on a different subset of topics or with different topic compositions) while committing to using one of these in practice.
> * Continuous ambiguity: The firm states for each topic a range for its possible importance in contributing to the total score.
>
> Since the uncertainties above are controlled by the firm, they are not attached to any intrinsic probabilities. Any (subjective) beliefs that users might hold would therefore be false, and work against them if they choose to respond accordingly, in particular since this lacks to account for the fact the firm is strategic in its choice of the realized model (and ambiguity set).

---

> > ### Author Rebuttal · Reviewer_Amho · 2026-03-31
> >
> > I thank the authors for their clear explanations and responses to my questions. My concern regarding the practicality of ambiguity has been addressed. I do now see the difficulty in comparing ambiguity with uncertainty performance wise. Perhaps this can be better explained in the camera ready version. Furthermore, the example with the offset, discrete, and continuous ambiguity was clear and informative.
> >
> > Therefore, I believe this paper should be accepted and will maintain my score. I think this is a good paper and a meaningful contribution to strategic classification.

---

> > > ### Author Response · Authors · 2026-04-06
> > >
> > > Thank you for your encouraging remarks! We appreciate your feedback and reviewing efforts, which have helped improve our paper.

---

### Official Review · Reviewer_qDRz · 2026-03-10

**Soundness:** 3
**Presentation:** 3
**Significance:** 2
**Originality:** 3
**Overall Recommendation:** 4
**Confidence:** 3

**Summary:**

This paper introduces the problem of ambiguous strategic classification; strategic from the fact that agents can modify their features (at a cost), ambiguous from the fact that the “system” can choose to disclose what family of classifiers it will select from rather than the exact classifier it will use. This leads to a dual task of learning both the “ambiguity set” to disclose and the exact classifier from that set to use. The objective is maximizing expected strategic accuracy for a randomly sampled agent. The specific cost model is the L2 norm between original and modified features, and the class of classifiers considered is the family of linear classifiers. Three ambiguity sets are considered: one where the weights w are set and the additive b term is disclosed to fall within a range (offset), one where multiple sets of (w,b) are provided (discrete), and one where each w_i falls within a disclosed range and the b term is fixed (continuous). For each one, a method of calculating the agents best responses are provided. In order to learn the ambiguity set and true classifier together, a loss function is provided, as well as an approximation of it that is tractable to optimize. Experiments are provided demonstrating that the size of the ambiguity set matters, and in fact a larger ambiguity set is not optimal.

**Compliance With Llm Reviewing Policy:**

Affirmed.

**Final Justification:**

I appreciate the time the authors took in generating both new theoretical results and experiments. They answer one of my main concerns regarding the polytope approximation for their strategic loss. It is especially nice that their worst case performance matches strategic hinge loss. While I am a bit uncertain on how to interpret the experimental results (what is a good \bar{R}?), I am happy to bump my score up to weak accept.

**Key Questions For Authors:**

1. What is the motivation, practically or theoretically, for the three ambiguity sets studied?
2. Do you have any intuition for what other forms of data would allow baseline unseparable data to become separable?
3. Are there existing methods that account for agent behavior, and if so how do those compare experimentally with yours?
4. How do the optimal agent behavior methods' time complexity scale with d?

**Limitations:**

Yes

**Strengths And Weaknesses:**

Strengths:
- This is an interesting setting and well-motivated
- I like the geometric interpretation of the problem, this provides a lot of room for applying geometry techniques in a new way
- The fact that ambiguity sets can cause typically unseparable data to become separable is very compelling, suggesting that leveraging strategic behavior (which usually makes things more challenging) can make certain tasks easier and even possible
- The loss function has a nice interpretation of projecting onto a set

Weaknesses:
- Agents being risk averse means they only move if this guarantees positive classification, which is very extreme behavior
- The ambiguity sets studied are very restricted: just linear and with very specific types of ambiguity
- The theoretical results in preliminaries and computing best responses sections are very simple, using very elementary linear optimization techniques
- No guarantees are provided for the loss function or the approximation guarantees of the outer polyhedral approximation
- The first separable case experiment feels designed to do well, rather than showing the effectiveness of the algorithm in a real setting
- The real data experiment achieves pretty low test accuracy, and comparing against a benchmark that completely ignores strategic action feels unfair

---

> ### Author Rebuttal · Authors · 2026-03-31
>
> Thank you for your careful reading and extensive review, we appreciate your positive thoughts on our proposed setting and approach. Please see below our response to your comments and questions, which we hope will address your concerns.
>
> > Agents being risk averse … is very extreme behavior
>
> Our setup captures settings in which users are willing to invest effort in order to guarantee outcome certainty. Consider our example of an important job interview exam. Candidates know the set of possible topics, but may not know which subset of topics will appear (discrete ambiguity), their relative importance in the final score (continuous ambiguity), or the decision threshold (offset ambiguity). We think it reasonable that in this case candidates will invest effort and study for all topics to ensure they pass the realized exam.
>
> > The ambiguity sets studied are very restricted: just linear and with very specific types of ambiguity
>
> Learning with ambiguity requires imposing some form of structure. We propose three distinct structure types that are simple yet natural, and capture realistic use cases for ambiguity; see motivation below. We believe that as a first paper on ambiguous strategic learning, these are a reasonable place to start.
>
> We focus on linear classifiers since this is by far the most common in strategic classification, and the only class for which, to our knowledge, a practical strategic learning algorithm exists.
>
> > The theoretical results in preliminaries and best response sections are very simple
>
> Our theoretical results are intended to provide readers with a general understanding of how ambiguity affects learning; we agree they are basic, and explicitly frame them as such (ln 142). We also agree that best responses under discrete ambiguity can be solved via reduction to convex (though not linear) programming—but do not see why this is a limitation, especially given that *learning* under such strategic behavior is not straightforward. As for computing continuous best responses, we hope you agree this is neither immediate nor elementary.
>
> > No guarantees are provided for the loss function
>
> The loss function and corresponding learning objective are non-convex. Providing convergence guarantees of e.g. SGD for this particular objective is an interesting question but is outside our scope.
>
> > The first separable case experiment feels designed to do well
>
> The first example uses a synthetic setting in which ambiguous learning *can* achieve significant and meaningful improvement to show that our method *does* in fact attain the theoretical optimum. We believe this a reasonable (and common) way to use synthetic data initially, with more realistic settings pursued in the following experiments.
>
> > The real data experiment achieves pretty low test accuracy comparing against a benchmark that completely ignores strategic action
>
> **Please note that the baseline *is* strategic**: it comprises an optimal strategic (non-ambiguous) classifier, and wraps it with a heuristically-defined ambiguity set. That is, it is strategic w.r.t. the learned classifier, but non-strategic w.r.t. ambiguity. This constitutes a sensible and natural baseline to compare against. Overall test accuracies are typical for this dataset—see Hardt et al. (2021).
>
> > What is the motivation for the three ambiguity sets?
>
> Consider the task of designing a qualifying exam, e.g., academic or for job hiring. Features represent answer correctness on questions of different topics, and weights represent the relative importance of each topic to the total score.
> * Offset ambiguity corresponds to determining in advance the importance of each topic, but keeping the acceptance threshold ambiguous.
> * Discrete ambiguity corresponds to releasing a set of possible exams (e.g., each focusing on a different subset of topics) while committing to using one of these.
> * Continuous ambiguity corresponds to stating for each topic a range for its possible importance.
>
> > Do you have any intuition for what other forms of data would … become separable?
>
> Yes. For a model class H, the induced effective class consists of all classifiers whose positive region is a union of the realized h and an intersection of all other $h’ \in \Gamma$ (line 362 (right)). Any data that admits to this form is in principle ambiguously separable.
>
> > Are there existing methods that account for agent behavior? How do those compare experimentally with yours?
>
> The only practical learning method for (batch) strategic classification we are aware of is the strategic hinge, which assumes users have full information, and applies only to linear classifiers. We use this method as the basis for the baselines.
>
> > How do the optimal agent behavior methods' time complexity scale with d?
>
> The ellipsoid method is poly(d). Our cutting plane method is worst-case exp(d), but in practice converged after less than d iterations on average, and never required more than 2.5d steps. We provide runtime statistics in Appendix D.

---

> > ### Author Rebuttal · Reviewer_qDRz · 2026-04-03
> >
> > Thank you for your thorough response to my concerns and questions.
> >
> > Regarding the motivation of the ambiguity sets studied, I appreciate the framing of them in the context of the job interview and recommend including this in the paper to provide additional intuition.
> >
> > However, my core concerns remain. I still feel that the theoretical results, including Theorem 1 for continuous ambiguity, are elementary to the point of weakening the paper. I acknowledge that the techniques use convex optimization rather than linear, that was my mistake. Theorem 1 specifically feels like a simple application of cutting planes, and the reduction from Equation 7 to 9 follows naturally from standard arguments in combinatorial optimization.
> >
> > That being said, I believe the strongest contribution here is the loss function. As noted in my original review, formal analysis of the approximation quality of \Tilde{F} relative to F remains the key gap in that section and would substantially strengthen the paper. Are the authors able to provide such bounds?
> >
> > More broadly, my view remains that the paper either needs stronger theoretical guarantees, whether for the loss function or a broader class of ambiguity sets, or a stronger experiment section to justify the practical use of these algorithms.

---

> > > ### Author Response · Authors · 2026-04-08
> > >
> > > Thank you for your continued reviewing efforts, we appreciate your willingness to engage with us.
> > >
> > > Thank you also for clarifying your question on approximation—we now understand you were referring to the relation between $\tilde{F}$ and$F$ as it affects the loss. As per your request, please find below (i) an initial theoretical analysis, and (ii) a complementing empirical analysis suggesting that in practice it is often tight.
> > >
> > > **Theory:**
> > >
> > > Given $h$ and $\Gamma \ni h$, the positive region $F(x) \ge 0$ can be defined as $A \cup B$ where $A$ is the positive region of $h$ and $B$ is the Minkowsky sum of $AOP(\Gamma)$ and an $\ell_2$ ball of radius $2/\alpha$. Since the AOP is a polytope, $B$ is an expanded, “rounded” polytope with ball segments instead of vertices. The approximation $\tilde{F}$ outer-bounds these “corner” ball segments by extending the faces of the (rounded) polytope to obtain a standard polytope. Note our assumption on classifier agreement implies ball segments are restricted to an orthant.
> > >
> > > Denote the true and approximate ambiguous losses by $L^\*$ and $\tilde{L}$. Define the distortion as $R = |\tilde{L} - L^*|$. We begin with some observations, and then provide a formula for $R$.
> > >
> > > Consider a point $x$, and denote by $x’$ its projection onto $F$ in the direction of $w$. First, notice that $\tilde{F}$ and $F$ agree on all the faces of $\tilde{F}$. This implies that **$\tilde{L}$ might be approximate only if $x’$ lies on a ball segment**. Conversely, if $x’$ lies on a face of $B$ or on $A$, then $\tilde{L}$ is exact $R=0$. We expect this to be the common case, as in our experiments below.
> > >
> > > We next provide an upper bound on $R$.
> > >
> > > **Claim**:
> > > Let $x$ with $x’=x+t^* w$ on a ball segment $C$ whose center $c$ is a vertex of the AOP. Define by $\beta$ the angle between the line from $c$ to $x’$ and the normal of $-w$. Then:
> > > $$|\tilde{L}-L^*| \le 2/\alpha \|\|w\|\|(1-cos(\beta))$$
> > > and
> > > $$\beta\le\max_{h_1,h_2 \in H(c)} 180-angle(h_1, h_2)$$
> > > Where $H(c)$ include all classifiers from $\Gamma$ intersecting at $c$.
> > >
> > > The bound is provably tight when the true $h \in H(c)$ and determines $\tilde{t}$.
> > >
> > > One observation is that the worst-case bound matches the distortion of the standard (non-ambiguous) strategic hinge. That is, **the worst-case ambiguous approximation is no worse than the alternative baseline**. This occurs only if (i) $c$ is an orthogonal vertex, and (ii) $x’$ lies near the boundary of $C$ (where it connects to a face). We expect this to be a rare event; in our experiments below, the average angle was 49.9 degrees.
> > >
> > >
> > > **Experiments:**
> > >
> > > For concreteness we focus on our Gaussian data experiment. Below we detail the procedure and provide a table with all results.
> > >
> > > *Exactness.*
> > > As noted, the approximation of $x$ is exact if its directional projection $x’$ lies on a face of $B$ or on $A$. This definition is intuitive, but not easily verified. Instead, we propose an equivalent procedural definition that is computationally feasible.
> > >
> > > Given $x$:
> > > 1. Compute $\tilde{t}$. If it derives from the outer min, return EXACT.
> > > 2. Compute $x’$ and its projection to the AOP. If only one constraint is tight, return EXACT.
> > > 3. Return APX.
> > >
> > > Note all steps above are implementable via the tools we provide in the paper.
> > >
> > >
> > > *Empirical distortion.*
> > > We next examine $\bar{R}$ for non-exact points. Note this requires computing $t^\*$. For this, we use two properties: (i) the distance of points $z$ on the line between $x$ and $x’$ to the center $c$ of $C$ is convex in $z$, and (ii) $t^\*$ is exactly at distance $2/\alpha$ from $c$. Together, these imply we can use line search to find $t^*$.
> > >
> > > Importantly, we would like to emphasize again that the *ability* to compute $t^\*$ (for reasonably sized sets) does not imply that *learning* with $t^*$ is tractable.
> > >
> > >
> > > *Results*.
> > > We applied the above procedure to all points across all experimental conditions. For scaling we report the normalized distortion $\bar{R}=|\tilde{L}-L^\*| / \bar{L^\*}$ where $\bar{L^\*}$ is the average true loss.
> > >
> > > On average, 55% of points were exact. For non-exact points, the average $\bar{R}$ was 0.13, and the 95th percentile was 0.61. In detail:
> > >
> > > | d | k | % APX | avg | 95% |
> > > |---|---|------|--------|-----|
> > > | 2 | 2 | 13% | 0.07 | 0.18 |
> > > | 2 | 3 | 19% | 0.07 | 0.26 |
> > > | 2 | 4 | 29% | 0.08 | 0.33 |
> > > | 2 | 5 | 53% | 0.11 | 0.42 |
> > > | 2 | 6 | 59% | 0.12 | 0.52 |
> > > | 2 | 7 | 54% | 0.11 | 0.49 |
> > > | 3 | 2 | 7%  | 0.07 | 0.36 |
> > > | 3 | 3 | 27% | 0.08 | 0.22 |
> > > | 3 | 4 | 45% | 0.10 | 0.44 |
> > > | 3 | 5 | 64% | 0.16 | 0.81 |
> > > | 3 | 6 | 67% | 0.20 | 1.00 |
> > > | 3 | 7 | 71% | 0.25 | 1.35 |
> > > | 4 | 2 | 19% | 0.07 | 0.21 |
> > > | 4 | 3 | 57% | 0.09 | 0.31 |
> > > | 4 | 4 | 68% | 0.17 | 0.91 |
> > > | 4 | 5 | 66% | 0.19 | 1.00 |
> > > | 4 | 6 | 73% | 0.25 | 1.34 |
> > > | 4 | 7 | 53% | 0.18 | 0.89 |
> > > | 5 | 2 | 7%  | 0.05 | 0.20 |
> > > | 5 | 3 | 26% | 0.07 | 0.25 |
> > > | 5 | 4 | 32% | 0.10 | 0.38 |
> > > | 5 | 5 | 51% | 0.13 | 0.67 |
> > > | 5 | 6 | 60% | 0.18 | 0.98 |
> > > | 5 | 7 | 63% | 0.19 | 1.04 |
> > > |   | average | 45% | 0.13 | 0.61 |

---

### Official Review · Reviewer_wiL4 · 2026-03-11

**Soundness:** 4
**Presentation:** 4
**Significance:** 3
**Originality:** 3
**Overall Recommendation:** 5
**Confidence:** 4

**Summary:**

The paper studies a strategic classification problem where agents manipulate their features at an $\ell_2$​ cost, but do not observe the deployed classifier exactly and instead respond conservatively to a set of possible linear classifiers of the form $w^Tx+b$. The paper considers three ambiguity models: (1) discrete ambiguity, where the agent faces a finite set of possible classifiers, (2) offset ambiguity, where w is fixed and the intercept b varies, and (3) continuous ambiguity, where the intercept b is fixed and w varies. The main contributions are: (1) introducing this ambiguous strategic classification model, in which agents manipulate only when they can guarantee acceptance across the ambiguity set, (2) characterizing the induced strategic responses and showing that a nontrivial amount of ambiguity can improve over full disclosure, and (3) proposing a practical optimization approach and empirical evaluation on synthetic data and the folktables dataset for learning such ambiguous classifiers.

**Compliance With Llm Reviewing Policy:**

Affirmed.

**Key Questions For Authors:**

This may be out of scope for the paper. I suppose the learner knows the cost function, and in particular the tradeoff parameter alpha. Do you have intuition regarding the result without knowing this parameter or what would happen if different users have different alpha’s?
Do you have any intuition regarding extending the results to more general cost classes (even l_p) and/or hypothesis class with finite vc dimension?
Any intuition regarding what happens when the ambiguity set is defined using both w and b?

**Limitations:**

yes

**Strengths And Weaknesses:**

*Strengths*

- The paper studies an interesting extension of strategic classification, where uncertainty about the deployed classifier is modeled through ambiguity sets and agents respond according to a worst-case acceptance criterion. This moves beyond Bayesian formulations, such as Cohen et al. (2024b), avoiding the need for a common prior over classifiers.
- The formulation is clear and, overall, the paper is well written.
- I find Theorem 1 not trivial. In the continuous ambiguity model, feasibility is defined with respect to a continuum of possible classifiers, so it is not obvious a priori that the projection problem admits an efficient finite characterization. The theorem establishes the correctness of Algorithm 1 by showing that the true projection can be recovered from a finite collection of active extreme constraints.
- The optimization discussion in Section 5 is also a nice part of the paper. Once agents best-respond to a set of classifiers under a worst-case criterion, the learning problem becomes substantially less standard than ordinary strategic ERM, so providing a tractable surrogate and an implementable training procedure adds real value to the paper.

*Weaknesses*
-  The scope of the model is somewhat narrow, as the analysis is restricted to linear classifiers and l2 manipulation costs. While this is common in the strategic classification literature, it still limits the conceptual reach of the results.
- The paper considers offset ambiguity and continuous ambiguity separately, but does not seem to treat joint ambiguity over both the linear coefficients w and the intercept b simultaneously. It would be helpful to explain whether this is mainly for technical convenience, whether the joint case becomes qualitatively different, or whether the current methods could extend to that setting.
- Some of the results would benefit from more proof intuition/idea/sketch in the main text.
- The assumption of the continuous ambiguity set in section 5 seems somewhat ad hoc. The paper assumes that all candidate classifiers should satisfy a weak-agreement condition and implements this via an additional penalty term, but I was missing a stronger justification for why this is the appropriate constraint and how important it is in practice for the method’s behavior.

*Comments*


- Burden definition should have ∣{i:y_i​=1}| instead of ∣{y_i​=1}|
- Lemma 1 should be an observation.
- Figure 1 would benefit from a more detailed caption. In particular, the meaning of the arrow colors and the different regions should be explained directly in the caption so that the figure is easier to read independently of the main text.
- Some of the lemma statements are not sufficiently self-contained. For example, symbols such as $\Omega$ and $\Gamma$ are introduced without immediately recalling their meaning.
- I think the threshold interpretation of offset ambiguity should be emphasized more explicitly. Interpreting this model as uncertainty over the effective decision threshold makes the setup more intuitive, and would also help make results such as Lemma 2 easier to parse.

Overall, I enjoyed reading this paper and recommend acceptance.

---

> ### Author Rebuttal · Authors · 2026-03-31
>
> Thank you for your positive review and encouraging feedback! Please see below our response to your questions and comments.
>
> > The scope of the model is somewhat narrow, as the analysis is restricted to linear classifiers and l2 manipulation costs. While this is common in the strategic classification literature, it still limits the conceptual reach of the results.
>
> Please note that **our results hold more generally**, in particular:
> * Our best-response algorithm for discrete ambiguity (Eq. (6)) applies to any convex cost function and to any class $H$ of convex classifiers.
> * Our best-response algorithm for continuous ambiguity (Algo. 1) applies to any convex cost function.
> * Our learning approach in Sec. 5.1 extends to any norm-based cost of the form $c(x,x’)=\phi(\|\|A^{-1/2} (x-x’)\|\|)$ where $\|\| \cdot \|\|$ can be any $p$-norm ($p \ge 1$), $\phi$ is a non-decreasing function, and $A$ is a PD matrix. This follows from the same procedure as [1], essentially replacing \|\|w\|\| in the hinge loss reward term $r$ with the dual $\|\|A^{-1/2} w\|\|_*$.
> * Both best-response and ambiguous hinge extend to simultaneous continuous and offset ambiguity (more details below).
>
> We will gladly revise the paper to reflect the most general case each of our results covers.
>
> We focus on linear classifiers since, to the best of our knowledge, this is the only class for which practical strategic classification algorithms exist. Our choice to focus on L2 costs was mostly to simplify writing and improve clarity, but we will gladly describe the most general cases which our results cover in the paper.
>
>
> > The paper considers offset ambiguity and continuous ambiguity separately, but does not seem to treat joint ambiguity over both the linear coefficients w and the intercept b simultaneously. It would be helpful to explain whether this is mainly for technical convenience, whether the joint case becomes qualitatively different, or whether the current methods could extend to that setting.
>
> This is certainly possible, and all relevant results (Lemma 5, Algo. 1, and Eq. (15)) easily extend to this joint case by simply replacing $b$ with $\underline{b}$, just as in the offset case in Sec. 4.1. Our choice to separate them was done mostly for clarity, as adding ambiguity to the offset on top of weights provides only mild further expressive power, and of the same form as offset ambiguity alone. For our experiments we found this useful as it enabled isolating the effect of weight ambiguity on performance, but if you feel this is important we will gladly add a baseline that learns both types jointly.
>
>
> > Some of the results would benefit from more proof intuition/idea/sketch in the main text.
>
> Thank you for your suggestion, we will gladly add these in the final version using the extra space.
>
>
> > The assumption of the continuous ambiguity set in section 5 seems somewhat ad hoc. The paper assumes that all candidate classifiers should satisfy a weak-agreement condition and implements this via an additional penalty term, but I was missing a stronger justification for why this is the appropriate constraint and how important it is in practice for the method’s behavior.
>
> The motivation behind the agreement requirement is that it ensures any movement made in response to one classifier improves the moving point’s score w.r.t. all other classifiers as well. We think that normatively this is a desirable property, as it restricts the learner’s ability to release contradicting classifiers (the extreme version of which is to release $h$ and $1-h$, which suppress movement entirely). It is also weaker (in some sense) that other possible normative constraint, for example demanding agreement in accuracy or in predictions. Technically, this assumption guarantees the correctness of the closed form solution for $\tilde{t}$ (below Eq. (15)).
>
>
> > Comments
>
> Thanks for these, we will fix the issues mentioned and improve clarity. We will elaborate on the threshold interpretation, as well as add further motivation for the other ambiguity types.
>
> [1] Rosenfeld & Rosenfeld. One-shot strategic classification under unknown costs. ICML 2024.

---

> > ### Author Rebuttal · Reviewer_wiL4 · 2026-04-01
> >
> > I thank the authors for their detailed rebuttal. It has strengthen the overall impact of the work. I am happy to maintain my score in support of acceptance.

---

> > > ### Author Response · Authors · 2026-04-06
> > >
> > > Thank you! We are happy our response was helpful. We appreciate your constructive feedback, and will implement in the paper all points raised.

---

### Official Review · Reviewer_GSVR · 2026-03-17

**Soundness:** 3
**Presentation:** 2
**Significance:** 2
**Originality:** 3
**Overall Recommendation:** 4
**Confidence:** 3

**Summary:**

In this paper, the authors propose the setting of ambiguous strategic classification. Specifically, the learner reveals a set of possible classifiers to the users, and the users aim to optimize their worst-case outcome when making strategic deviations. Additionally, the authors consider three specific types of ambiguity sets in linear classification settings. To address this problem, they first analyze the best responses of users facing ambiguous sets and then study how the learner should optimize accordingly. Experiments on both synthetic and real data show that the proposed method can achieve better performance than the baseline methods.

**Compliance With Llm Reviewing Policy:**

Affirmed.

**Final Justification:**

I would like to raise my score to 4.

**Key Questions For Authors:**

See the weakness part.

**Limitations:**

yes

**Strengths And Weaknesses:**

**Strengths**:

1. The ambiguous strategic classification setting is novel and interesting.
2. The paper provides useful intuition and insights into how users may behave and how learners should optimize in the proposed setting.

**Weaknesses**:

1. Although the general setting is interesting, the paper only considers rather restricted concrete configurations, including the $2$-norm cost function, the linear classification setting, and three specific types of ambiguity sets. The proposed solution appears to depend heavily on these specific choices, and it is unclear how the approach could be extended to more general settings.
2. The motivation for the three specific types of ambiguity sets is not very clear. In addition, some of the technical difficulty of the paper lies in the continuous ambiguity set. However, I think a more natural way to model the continuous $w$ setting would be to borrow ideas from robust linear programming, where the uncertainty set is assumed to be an ellipsoid. In that case, the resulting optimization problem might possibly be solved via another convex program.
3. I do not understand why the third graph in Figure 1(B) is labeled as having "no burden." It seems that users in the shaded yellow region could still incur social burden, even if they are classified as positive by the true classifier. In addition, I do not follow why the example in Figure 1(C) leads to ambiguous separability. Could the authors provide more details on this example?
4. For the learning approach, it is not clear how the authors actually optimize both the ambiguity set and the classifier. It would be helpful if the authors could provide pseudocode for the proposed learning algorithm and discuss the optimization procedure in more detail, including how the approximation is carried out in practice.
5. There are also several unclear points in the writing. The sentence "Let $\Gamma$" is incomplete in Lemma 1, and it is also incomplete in Lemma 4. The notation $\Delta_h^{\Gamma}(h)$ in Equation (3) is confusing, since the right-hand side of the equation does not depend on $h$. The notation $[\underline{h}, \overline{h}]$ in Equation (7) is not well defined when $\underline{h}$ and $\overline{h}$ are vectors. The lines in Figure 1 are not well explained. The period should be replaced by a comma in the sentence "Second, under ambiguity, standard losses ..." in Line 82.

---

> ### Author Rebuttal · Authors · 2026-03-31
>
> Thank you for your detailed review and questions. Please see our response to your comments below. In particular, we include motivation for our studied ambiguity types, a description of the most general setting our results cover, and a detailed explanation of the optimization procedure. We hope these help address your concerns.
>
> > The paper only considers rather restricted concrete configurations: 2-norm cost, linear classifiers, and three specific types of ambiguity sets.
>
> We focus on linear classifiers since, to the best of our knowledge, this is the only class for which practical strategic classification algorithms exist.
> As for costs, we focused on L2 for simplicity and clarity. However, **please note that our results hold more generally**. In particular:
> * Our best-response algorithm for discrete ambiguity (Eq. 6) applies to *any convex cost* (and to any class H of convex classifiers).
> * Our best-response algorithm for continuous ambiguity (Algo. 1) applies to *any convex cost*.
> * Our learning approach in Sec. 5.1 extends to *any norm-based cost* of the form $c(x,x’)=\phi(\|\|A^{-1/2} (x-x’)\|\|)$ where $\|\| \cdot \|\|$ can be any $p$-norm ($p \ge 1$), $\phi$ is a non-decreasing function, and $A$ is a PD matrix.
>
> The latter follows from the same procedure as [1], essentially replacing $\|\|w\|\|$ in the hinge loss reward term $r$ with the dual $\|\|A^{-1/2} w\|\|_*$.
>
> We believed that focusing on the basic setting would simplify writing and improve clarity, but
> will gladly revise the paper to reflect the most general case each of our results covers.
>
> Regarding ambiguity types, please see below.
>
>
> > The motivation for the three specific types of ambiguity sets is not very clear.
>
> Consider for example the task of designing a qualifying exam, e.g., academic or for job hiring. Here features represent answer correctness on questions of different topics or subjects, and weights represent the relative importance of each topic to the total score.
> * Offset ambiguity corresponds to determining in advance the importance of each topic via a fixed score function, but keeping the acceptance threshold ambiguous.
> * Discrete ambiguity corresponds to releasing a set of possible exams (e.g., each focusing on a different subset of topics) while committing to eventually using one of these.
> * Continuous ambiguity corresponds to stating for each topic a range for its possible importance in the final weighted sum.
> We believe these are natural candidates for our initial study of ambiguity in strategic learning.
>
>
> > I think a more natural way to model the continuous setting would be [to assume] the uncertainty set is an ellipsoid.
>
> This is an interesting idea, though we are not sure what set of classifiers would give rise to this form of ambiguity, nor whether this would be qualitatively different from our definition of continuous ambiguity. However, it does sound like an interesting extension worth pursuing.
>
>
> > I do not understand why the third graph in Fig. 1(B) is labeled as having "no burden”.
>
> You are correct—apologies for the imprecise statement. We will fix the label to be “low burden” instead.
>
>
> > I do not follow why the example in Fig. 1(C) leads to ambiguous separability.
>
> The figure shows the AOP but not the set of classifiers that induce it—perhaps this would help clarify: Consider $\Gamma$ consisting of three classifiers: (i) horizontal just above the data, (ii) vertical just to the right of the data, and (iii) diagonal at the top-right corner of the positives. The maximal distance points move is the width of the grey positive region plus some small $\epsilon$. Thus, all positive points can cross all three classifiers, and are labeled positive after movement. However, negative points can cross either the horizontal or vertical classifiers, but not both, and so cannot move and hence are labeled negatively. We will add these elements to the figure.
>
>
> > For the learning approach, it is not clear how the authors actually optimize both the ambiguity set and the classifier.
>
> The general idea is to take gradients w.r.t. the parameters of all classifiers $h \in \Gamma$, where all possible classifiers $h’$ affect the reward term $r$ in the ambiguous hinge loss, but only the realized $h$ affects the penalty via the score $w^\top x + b$. The challenge is that $r$ itself depends on the argmin of a different optimization problem (Eq. 13). What we show is that this can be circumvented by taking the max, which is differentiable, over a set of candidate points which can be computed in closed form (see $\tilde{t}$ below Eq. 15). We will add pseudocode and a more detailed description of the procedure to the Appendix.
>
> > There are also several unclear points in the writing.
>
> Thank you for pointing these out! We will clarify all notations, fix the broken sentences, further elaborate on Fig. 1, and remove the subscript $h$ from $\Delta$.
>
> [1] Rosenfeld & Rosenfeld. One-shot strategic classification under unknown costs. ICML 2024.

---

> > ### Author Rebuttal · Reviewer_GSVR · 2026-04-02
> >
> > I appreciate the authors’ response. I have a further question about Figure 1(B). Is it correct that the designer uses the yellow lines to classify the samples? If so, then there seem to be some green regions that are still not classified as positive, even when strategic behavior is allowed. In that case, would there still be false negatives?

---

> > > ### Author Response · Authors · 2026-04-06
> > >
> > > Thank you for your feedback and your willingness to further engage with us.
> > >
> > > In Fig. 1 (B), points are classified according to the *purple* classifier (solid line) – this is the realized classifier $h \in \Gamma$. The yellow classifiers (solid lines) are the additional possible classifiers in the ambiguity set, $h_1, h_2 \in \Gamma$. These only affect how points move, and do not determine prediction outcomes directly. The dashed lines mark distance $2 / \alpha$ from each classifier. These notations follow those in Fig. 1 (A); apologies if the connection was not clear, we will make sure to point this out.
> > >
> > > The green (+) and blue (-) areas correspond to data regions with true labels $y =+1$ and $y=-1$, respectively. Please note that points in the small green “triangle” regions at the top and bottom (which we believe you are referring to) **are classified correctly** since they are already on the positive side of $h$ (purple solid line), and will remain so even if they do move.
> > >
> > > The more interesting points are those in the blue regions that are also shaded purple. These are negative points whose movement is suppressed due to ambiguity: they are too far from the AOP—i.e., the intersection of all $h’ \in \Gamma$ (purple and both yellows), which in this case is the large green triangle on the right. Since these points are initially on the negative side of $h$ *and* do not move, they will be correctly classified as negative. Thus, the data is fully separable using the ambiguous set $\Gamma  = \\{h, h_1, h_2 \\}$.
> > >
> > > We hope this addresses your question – we will make sure to revise the figure and caption to clarify this point.

---

### Decision · Program_Chairs · 2026-04-30

**Decision:**

Accept (regular)

**Comment:**

The paper studies strategic classification in a linear setting, where the agent responds conservatively to a collection of possible classifiers. All reviewers like the paper, identifying strengths such as the natural model, nontrivial technical contributions, and interpretations thereof. There were minor concerns, most of which were resolved after the author response phase. Overall I believe the paper would be a solid contribution to ICML.